# Identification of oxidative stress-responsive genes in recurrent miscarriage and their role in disease pathogenesis

**Mian Wang[☉], Lingling Zhu[☉], Xiaoyan Cheng[iD]***

Department of Obstetrics, Affiliated Maternity and Child Health Care Hospital of Nantong University, Nantong, Jiangsu, China

[☉] These authors are contributed equally to this work.
* 1012770174@qq.com

## Abstract

Recurrent miscarriage (RM) is a distressing reproductive condition affecting approximately 1–3% of couples. The underlying causes related to oxidative stress remain largely unclear and necessitate additional research. This study aimed to employ bioinformatics approaches to uncover the differential expression of oxidative stress-responsive genes in RM to elucidate their potential involvement in the disorder etiology. Upon examination of the data retrieved from the Gene Expression Omnibus (GEO), 18 oxidative stress-responsive differentially expressed genes (OSRDEGs) were identified. Bioinformatics techniques, which led to the identification of six hub genes—*ARRB2*, *BMF*, *SORCS2*, *STK3*, *UCN2*, and *VIPR1*—as potential biomarkers for RM, elucidated the biological processes and molecular functions associated with genes that are differentially expressed under oxidative stress. GSEA enrichment profiling revealed significant enrichment of genes between the RM and control groups in pathways such as hypoxia, epithelial–mesenchymal transition (EMT) in breast tumors, upregulation of Wnt signaling in liver cancer progenitors, and TGFβ-induced EMT. The mRNA-miRNA interaction network analysis revealed five hub genes interacting with 41 miRNAs, with *STK3* exhibiting the highest connectivity among miRNA interactions. Additionally, the analysis of immune cell infiltration demonstrated a substantial inverse relationship between *ARRB2* and the levels of plasma cells and neutrophils. Conversely, *UCN2* was positively associated with T. cells. CD8 are inversely associated with monocytes. Furthermore, immunohistochemical (IHC) analysis on endometrial tissues from RM patients and matched controls confirmed significantly elevated protein expression levels of six hub genes in the RM group, consistent with the bioinformatics findings. This study establishes a diagnostic model and provides insights into immune-modulation therapies for RM.

**Data availability statement:** All data relevant to the study are included in the article or available as supplemental information. Gene expression data are available in the GEO repository (https://www.ncbi.nlm.nih.gov/geo/) under accession numbers GSE22490, GSE26787, and GSE165004.

**Funding:** This work was supported by the Municipal Health Commission of Nantong (No. MSZ2023057). The funders had no role in study design, data collection and analysis, decision to publish, or preparation of the manuscript.

**Competing interests:** The authors have declared that no competing interests exist.

## Introduction

Recurrent miscarriage (RM) refers to the phenomenon of experiencing two or more sequential pregnancy losses before the 20th week of gestation, impacting approximately 1–3% of couples attempting to conceive [1,2]. Despite its common occurrence, the underlying causes of RM are not fully understood, and many cases remain unexplained, with no definitive solution currently available [3]. Therefore, there is an urgent need for improved diagnostic tools and treatment strategies to address these complex reproductive health issues.

Oxidative stress (OS) occurs when free radicals exceed the body's antioxidant capacity to neutralize them [4]. Research has shown that OS can lead to endothelial damage, compromised placental vasculature, immune dysfunction, and negative pregnancy outcomes [5]. It has also been implicated in various pregnancy-responsive disorders, including RM, preeclampsia, fetal growth restriction, preterm labor, and preterm premature rupture of membranes [6]. Most knowledge regarding the role of OS in pregnancy comes from animal studies. Ishii *et al*. [7] demonstrated that oxidative stress in the mitochondria could cause placental inflammation and halt embryonic development in mice. These findings suggest that reducing OS could be a potential therapeutic strategy for treating spontaneous RM.

Additionally, Wang *et al*. [8] used bioinformatic analysis to identify differentially expressed genes in endometrial samples from the RM and control groups, thereby pinpointing potential biomarkers for RM. However, previous studies have not explored the differential expression of genes related to RM and OS.

Within the scope of this study, we retrieved the GSE22490 and GSE26787 datasets from the Gene Expression Omnibus (GEO) database [9] and merged them into a combined dataset. We analyzed these datasets as a test set, with GSE165004 as the validation set. We comprehensively analyzed the oxidative stress-responsive differentially expressed genes (OSRDEGs) in RM by integrating data from multiple gene expression datasets. We scrutinized the variability in gene expression specific to RM and conducted analyses of functionality, pathways, and gene set enrichment. Furthermore, we developed machine learning algorithms to devise diagnostic models. In addition, we analyzed immune cell infiltration in RM samples to elucidate the interplay between OS and the immune response in this disease. This holistic methodology is expected to provide a novel understanding of the molecular underpinnings of RM and establish a foundation for innovative diagnostic and therapeutic approaches.

## Materials and methods

### Data download

We retrieved the datasets GSE22490 [10], GSE26787 [10], and GSE165004 [10] from patients with RM from the GEO database [9] via GEOquery [11]. All datasets were confirmed to be derived from *Homo sapiens*. The GSE22490 dataset consists of microarray gene expression profiles of the placenta, totaling 10 samples. Samples with > 12 gestational cycles were excluded. The resulting

dataset included three RM samples (RM group) and five normal samples (control group). These samples were analyzed via the Affymetrix Human Genome U133 Plus 2.0 Array, which was identified via the platform GPL570 [HG-U133_Plus_2]. The GSE26787 dataset consisted of microarray gene expression profiles of the endometrium, totaling 15 samples. This dataset included five RM samples and five normal samples. The analysis of these 10 samples was performed via the Affymetrix Human Genome U133 Plus 2.0 Array, which corresponds to the platform identifier GPL570 [HG-U133_Plus_2]. The GSE165004 dataset consists of microarray gene expression profiles of the endometrium, totaling 72 samples. This dataset included 24 RM and 24 normal samples. The comprehensive analysis of these 48 samples was executed on the Agilent-039494 SurePrint G3 Human GE v2 8×60K Microarray 039381 platform, identified under GPL16699. The GSE22490 and GSE26787 datasets were amalgamated to form the test cohort, and the GSE165004 dataset served as the validation cohort for further analysis. The specific dataset information is presented in Table 1.

We assembled a collection of oxidative stress-responsive genes (OSRGs) from the GeneCards database, a repository offering extensive information on human genes [12]. Using the term "oxidative stress" as a search keyword, we obtained 1558 OSRGs. We searched the Molecular Signatures Database [13] (MSigDB) website with "oxidative stress" as the key word, and 120 OSRGs were identified from the GObp Cell Death In Response to OS and Biocarta Arenrf2 Pathway reference gene sets. A total of 279 OSRGs were obtained from the literature [14]. Finally, the OSRGs from the three sources were merged with the combined datasets, and the genes of the GSE165004 dataset were intercrossed to obtain 1571 OSRGs, which were subsequently integrated into our analysis and categorized as OSRGs. The nomenclature of the genes is shown in S1 Table.

## Differentially expressed genes correlated with oxidative stress

Initially, we employed the sva R package [15] to perform batch correction on the GSE22490 and GSE26787 datasets, forming a combined datasets containing eight RM samples and ten control samples. Principal component analysis (PCA) [16] reduces data dimensions by extracting feature vectors from high-dimensional datasets and mapping them into a lower-dimensional space for visual representation in two- or three-dimensional graphs.

To discern differentially expressed genes (DEGs) associated with the RM and control groups, we utilized the limma package [17] for variance analysis of the combined datasets expression data. We set a threshold of |logFC|>0.5 and P.adjust<0.05 to identify DEGs, defining upregulated genes as those with logFC>0.5 and P.adjust<0.05 and downregulated genes as those with a logFC<−0.5 and P.adjust<0.05.

We then merged the datasets via variance analysis to identify genes with | logFC|>0.5 and P.adjust<0.05, linking them to 1571 OSRGs. By intersecting the datasets and constructing a Venn diagram, we identified the OSRDEGs. Subsequently, volcano plots and a differential ranking map were constructed via variance analysis via the ggplot2 R package [18], and a heatmap [19] was developed via the pheatmap R package.

**Table 1. GEO dataset information list.**

|  | GSE22490 | GSE26787 | GSE165004 |
|---|---|---|---|
| **Platform** | GPL570 | GPL570 | GPL16699 |
| **Species** | Homo sapiens | Homo sapiens | Homo sapiens |
| **Tissue** | placenta tissue | endometrium tissue | endometrium tissue |
| **Samples in Control group** | 5 | 5 | 24 |
| **Samples in RM group** | 3 | 5 | 24 |

RM, recurrent miscarriage.

## Functional enrichment analysis (GO) and pathway enrichment analysis (KEGG) between the RM group and the control group

Gene Ontology (GO) [20] serves as a standard method for conducting large-scale functional classification studies and includes the examination of biological processes (BP), molecular functions (MF), and cellular components (CC). The Kyoto Encyclopedia of Genes and Genomes (KEGG) [21] is recognized as an extensive database repository that archives data on genomes, biological pathways, diseases, and pharmaceuticals. Using the R package clusterProfiler [22], we performed GO and KEGG pathway analyses to identify OSRDEGs between the RM and control groups within the combined datasets. We deemed items with P.adjust < 0.05 and FDR value (q value) < 0.25 as statistically significant via the Benjamini–Hochberg (BH) correction approach.

## GSEA of the RM and control groups

Gene set enrichment analysis (GSEA) [23] was used to determine the influence of a preset gene collection on the phenotype by scrutinizing their arrangement in a ranked gene list correlated with phenotypic associations. The genes in the combined datasets were initially sorted and divided into high- and low-phenotypic-relevance groups. Next, we employed the clusterProfiler package to perform enrichment analysis comparing all genes between the RM and control groups. For the GSEA enrichment analysis, the following parameters were specified: a seed value of 2022, a calculation count of 1000, a minimum gene set size of 10, a maximum gene set size capped at 500, and BH correction for p values. We obtained the GMT gene set under c2.Cp.All.V2022.1. Hs. Symbol designation from the MSigDB repository as a reference. The thresholds for considering enrichment as significant were set at P.adjust < 0.05 and a q value < 0.05.

## Constructing a diagnostic model for RM

Initially, we employed the R package glmnet [24], utilized OSRDEGs, and set the number of seed parameters to 500. OSRDEGs were selected via least absolute shrinkage and selection operator (LASSO) regression, which enhances linear regression by incorporating a penalty term consisting of lambda multiplied by the absolute slope value to improve predictive accuracy. The outcomes of the LASSO regression were depicted via a diagnostic model and variable trajectory diagrams.

We developed a support vector machine (SVM) [25] model utilizing hub genes from the combined datasets, prioritizing genes with the highest precision and lowest error rate. The genes selected by the LASSO regression and SVM models were compared, and a Venn diagram was used to identify the overlapping genes related to necrotizing apoptosis. Additionally, the RCircos package [26] was used to visualize the chromosomal distribution of the hub genes.

Genes resulting from the overlap between the LASSO and SVM analyses were identified as hub genes for subsequent studies. The hub genes are shown in Table 2.

Semantic comparison within the GO annotation framework offers a numerical approach for assessing gene similarity, a critical foundation for numerous bioinformatics methodologies. Semantic similarity among hub genes was determined

**Table 2. List of hub genes from the differential expression analysis.**

| Gene Symbol | Description | logFC | P.Value | adj.P.Val |
|---|---|---|---|---|
| *UCN2* | urocortin 2 | 0.689570313 | 1.88E-06 | 0.002909085 |
| *VIPR1* | vasoactive intestinal peptide receptor 1 | 0.833457729 | 1.25E-05 | 0.009071088 |
| *BMF* | Bcl2 modifying factor | 1.125451843 | 0.00023692 | 0.028166077 |
| *STK3* | serine/threonine kinase 3 | 0.550161852 | 0.000679591 | 0.042964865 |
| *ARRB2* | arrestin beta 2 | 1.041823009 | 0.000693659 | 0.043239395 |
| *SORCS2* | sortilin related VPS10 domain containing receptor 2 | 0.544211573 | 0.000879962 | 0.046934934 |

via the GOSemSim package [27], with further computation of the geometric means of the BP, CC, and MF categories to establish the final score. Functional similarity analyses were performed via the ggplot package.

We visualized the outcomes of the functional similarity analysis via the ggplot package. To derive the OSRDEG diagnostic model within the combined datasets, we subsequently conducted logistic regression analysis focusing on the hub genes. Logistic regression was applied to the hub genes to develop a diagnostic model for OSRDEGs across the combined datasets. Regression was used to examine the binary relationship between the independent and dependent variables, which corresponded to the RM and control groups, to construct a logistic model. The RM samples were bifurcated into a high predictive value group (High) and a low predictive value group (Low) on the basis of the median predictive value derived from the logistic regression. In addition, a nomogram [28], a graphical tool that delineates the functional interplay between multiple independent variables within a rectangular coordinate system with disjoint line segments, was generated via the RMS package to indicate the relationships between hub genes within the logistic regression model.

Finally, a calibration analysis was conducted to produce a calibration curve, which served to evaluate the precision and discriminatory power of the logistic regression model developed from the hub genes. Decision curve analysis (DCA) [29] is a straightforward approach for appraising clinical forecasting models, diagnostic assays, and molecular markers. The ggDCA R package was used to generate a DCA plot, which facilitated the assessment of the model's accuracy and discrimination.

## Correlation analysis of the hub genes

We employed the Spearman method to assess the correlation between the hub gene expression levels, and the findings were depicted via a correlation heatmap.

## GSEA between the high and low groups

Initially, we categorized the genes within the combined datasets into two groups on the basis of their correlation with the phenotypic traits. The clusterProfiler package was subsequently used to perform enrichment analysis on the genes distinguishing the high- and low-phenotypic-correlation cohorts. In setting up the GSEA parameters, we designated the seed as 2022, established 1000 as the number of computations, set the threshold for genes per gene set at a minimum of 10 and a maximum of 500, and applied the BH method for p value correction. Our reference gene set was sourced from the MSigDB database under the GMT collection (c2.Cp.All.V2022.1.Hs. Symbols), with enrichment significance defined as P.adjust < 0.05 and a q value < 0.05.

## Construction of mRNA–miRNA and mRNA–transcription factor networks

The miRDB database [30] (http://mirdb.org) is an online tool for predicting functional miRNA targets. We used the miRDB database to predict interactions between miRNAs and hub genes and subsequently constructed an mRNA–miRNA network via Cytoscape software.

The CHIPBase repository (edition 2.0), accessible via its secure web address (https://rna.sysu.edu.cn/chipbase/) [31], employs ChIP-seq data of DNA-binding proteins to discern numerous composite base sequence patterns and their respective binding locations. Furthermore, it projects an extensive array of regulatory interactions between factors (TFs) and their target genes. The HTFtarget database [32], which can be accessed at http://bioinfo.life.hust.edu.cn/hTFtarget, is a holistic resource that encompasses information on human transcription factors and their targeted regulatory mechanisms and presents a detailed, trustworthy, and user-friendly environment for exploring the regulatory control exerted by human TFs. We explored the CHIPBase and HTFtarget databases to identify TFs that bind to hub genes and depicted their interactions via Cytoscape software.

## Analysis of immune cell infiltration (CIBERSORTx)

To assess the distribution of various immune cell types within RM samples, we used the analytical tool CIBERSORTx [33]. We synthesized gene expression datasets from the CIBERSORTx online resource (http://cibersortx.stanford.edu/) and cross-referenced them with the LM22 gene signature to construct a matrix indicating the extent of immune cell infiltration. A filter was then applied to retain only data points with immune cell enrichment scores exceeding zero, culminating in a matrix that specified distinct patterns of immune cell infiltration. The disparities in immune cell infiltration between the high- and low-RM sample groups were visually represented via stacked bar charts, as interpreted via logistic regression analysis. The interdependencies among the different immune cells within the RM samples were evaluated via Spearman's correlation coefficient and graphically displayed via the ggplot2 package in R. By integrating the gene expression data from the combined datasets, we ascertained the correlation between immune cells and hub genes in the stratified groups, and Spearman's correlation analysis was employed to interpret and visualize these correlations in a heatmap.

## Immunohistochemical detection of hub proteins

Endometrial tissue samples were obtained via hysteroscopic biopsy during the luteal phase from three patients diagnosed with RM and three matched control subjects. All tissue samples were fixed in formaldehyde and embedded in paraffin for sectioning. Following deparaffinization and rehydration, antigen retrieval was conducted on the paraffin-embedded tissue sections via citrate buffer (pH 6.0). Endogenous peroxidase activity was inhibited by incubation with 3% $H_2O_2$, and nonspecific binding was blocked with normal goat serum. The sections were incubated overnight at 4°C with primary antibodies, including BMF (Abcam, UK) as well as SORCS2, STK3, VIPR1, UCN2, and ARRB2 (all from Proteintech, USA). This was followed by incubation with an HRP-conjugated secondary antibody at room temperature for 30 minutes. Immunoreactivity was visualized using diaminobenzidine (DAB), and the nuclei were counterstained with hematoxylin. Two blinded pathologists independently evaluated all the slides via a semiquantitative H-score system. The scoring criteria were defined as follows: three points for strong positivity, two points for moderate positivity, one point for weak positivity, and zero points for negative staining.

The study protocol was approved by the Institutional Research Ethics Committee of Affiliated Maternity and Child Health Care Hospital of Nantong University. All procedures were conducted in accordance with the ethical principles outlined in the Declaration of Helsinki.

## Data analysis and interpretation

With R software version 4.3.1, we executed all the statistical computations and analytical interpretations for this study. We used the classic Student's t test for datasets that adhered to a Gaussian distribution to evaluate the statistical significance of the continuous data across the two groups. For datasets that were not normally distributed, we opted for the Mann–Whitney U test, a nonparametric alternative. In instances of data analysis involving three or more groups, we relied on the Kruskal–Wallis test as our primary analytical tool. We employed either the chi-square test or Fisher's exact test to determine significant disparities between the two groups. Spearman's rank correlation was used to determine the correlation coefficients between the various molecular entities. Unless stated otherwise, all P values were calculated on a two-tailed basis, with statistical significance indicated by a threshold of less than 0.05.

## Results

### Technology roadmap

To systematically investigate the role of OSRGs in RM, we designed a comprehensive analytical workflow (Fig 1). Our approach began with the retrieval and processing of gene expression data from multiple GEO datasets, followed by the identification of DEGs and OSRGs. Through the intersection of these gene sets, we identified OSRDEGs. We

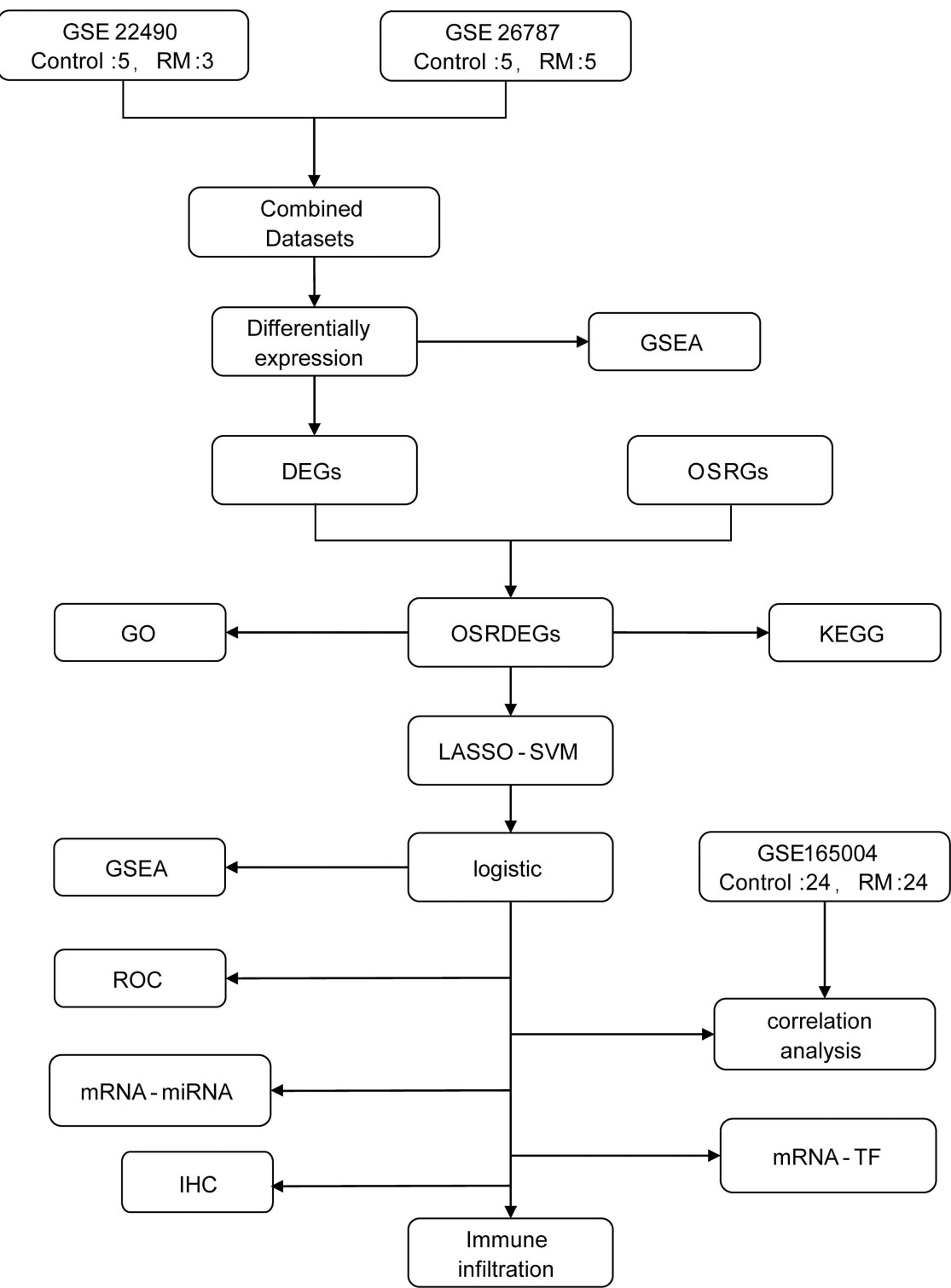

**Fig 1. Technology roadmap.** DEGs, differentially expressed genes. OSRGs, oxidative stress-responsive genes. OSRDEGs, oxidative stress-responsive differentially expressed genes. GO, Gene Ontology. KEGG, Kyoto Encyclopedia of Genes and Genomes. GSEA, Gene Set Enrichment Analysis. LASSO, least absolute shrinkage and selection operator. SVM, support vector machine. TF, transcription factor. ROC, receiver operating characteristic. IHC, immunohistochemistry. RM, recurrent miscarriage.

subsequently conducted extensive bioinformatics analyses, including GO and KEGG pathway enrichment, GSEA, and the construction of diagnostic models via LASSO regression and SVM. Additionally, we explored transcription factor networks and performed immune cell infiltration analysis to gain deeper insights into the molecular mechanisms underlying RM. To further elucidate the underlying molecular mechanisms, we conducted an integrated analysis of transcription factor regulatory networks and immune microenvironment infiltration. Ultimately, immunohistochemical(IHC) analysis was performed to validate the protein-level expression of pivotal candidate genes, thereby providing critical experimental corroboration for our bioinformatic predictions.

## Data collection and correction

Initially, we employed the sva R package to mitigate the influence of batch effects from the integrated GSE22490 and GSE26787 datasets, yielding a unified GEO datasets. Comparative visualizations were subsequently generated (Fig 2A and 2B) to illustrate the state of the datasets before and after batch effect mitigation. PCA plots, as depicted in Fig 2C and 2D, demonstrated consistency across the postcorrection datasets. Distribution boxplot and principal component analysis (PCA) plot analyses confirmed that the batch effect was substantially diminished after removal from the combined datasets.

## Oxidative stress-responsive differentially expressed gene analysis

To identify the OSRDEGs, we segregated the dataset into RM and control groups. Following this categorization, meticulous differential gene expression analysis was subsequently performed. The analysis identified 290 genes that surpassed the predefined thresholds for differential expression, with a notable |logFC| > 0.5 and P.adjust < 0.05. Among these genes, 151 were upregulated in the RM group relative to the control group, as indicated by logFC > 0.5 and P.adjust < 0.05. In contrast, 139 genes were downregulated in the RM group compared with the control group, as indicated by logFC < 0.5 and P.adjust < 0.05. To visualize the outcomes of the gene expression analysis, a volcano plot was constructed (Fig 3A), providing a clear illustration of the dataset's differential analysis. When genes whose expression was modulated by OS were targeted, the initial phase was the amalgamation of datasets from combined datasets. Following integration, meticulous differential expression analysis was conducted, applying the thresholds of |logFC| > 0.5 and P.adjust < 0.05. This process led to the identification of 290 genes whose expression significantly differed. The intersection of these genes with a pool of 1571 OSRGs yielded a subset of OSRDEGs that received 18 OSRDEGs (*ADM2, AOC3, APLN, ARRB2, BMF, CD40, GADD45G, GPT, IKBKE, IL18, MMACHC, MSRB3, NPPA, SORCS2, STK3, TREM2, UCN2, VIPR1*) and a Venn diagram (Fig 3B). On the basis of the intersecting data, we assessed the divergent expression profiles of OSRDEGs between the RM cohort and control subjects across various samples within the combined datasets. This led to the development of a heatmap (Fig 3D) and comparative graphical representation (Fig 3F) to illustrate the expression trends of OSRDEGs. Parallel to this approach, an analysis of gene expression variance was performed on the distinct samples of the GSE165004 dataset, RM versus control, resulting in the depiction of a heatmap (Fig 3E) and a comparative chart (Fig 3G) for the OSRDEGs. Gene expression analysis revealed that the genes *GADD45G, IKBKE, MMACHC,* and *MSRB3* were markedly distinct across both the combined and GSE165004 datasets. To capstone the analysis, the chromosomal positioning of 18 OSRDEGs was examined via the RCircos R package, culminating in the visualization of a chromosome distribution map (Fig 3C). Chromosome mapping revealed that OSRDEGs were generally located on chromosomes 1, 3, 4, 6, 8, 9, 11, 12, 15, 17, 20, 22 and X.

## GO and KEGG profiling of OSRDEGs in the RM and control groups

To elucidate the BPs, MFs, and CCs of the 18 OSRDEGs in the RM and control groups, we performed a comprehensive analysis. The analysis included GO enrichment to shed light on the roles of genes and a KEGG pathway study to chart the pathways linked to OSRDEGs. Fig 4A presents the results of both the GO functional and KEGG pathway enrichment

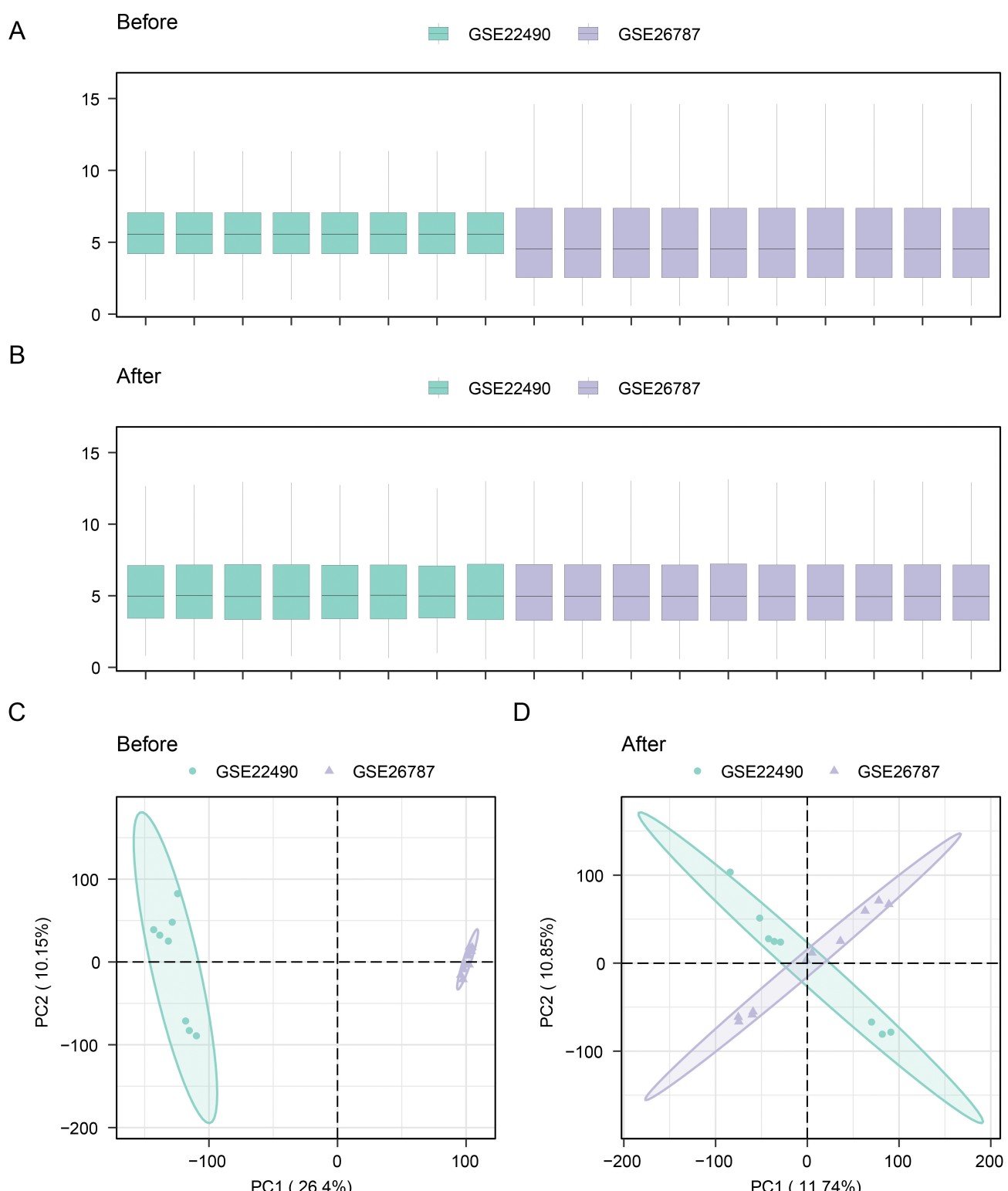

**Fig 2. Data collection and batch effect correction. A.** Distribution boxplot of the dataset before batch effect correction. **B.** Distribution boxplot of the dataset after batch effect correction. **C.** PCA plot of the dataset before batch effect correction. **D.** PCA plot of the dataset after batch effect correction. PCA, principal component analysis.

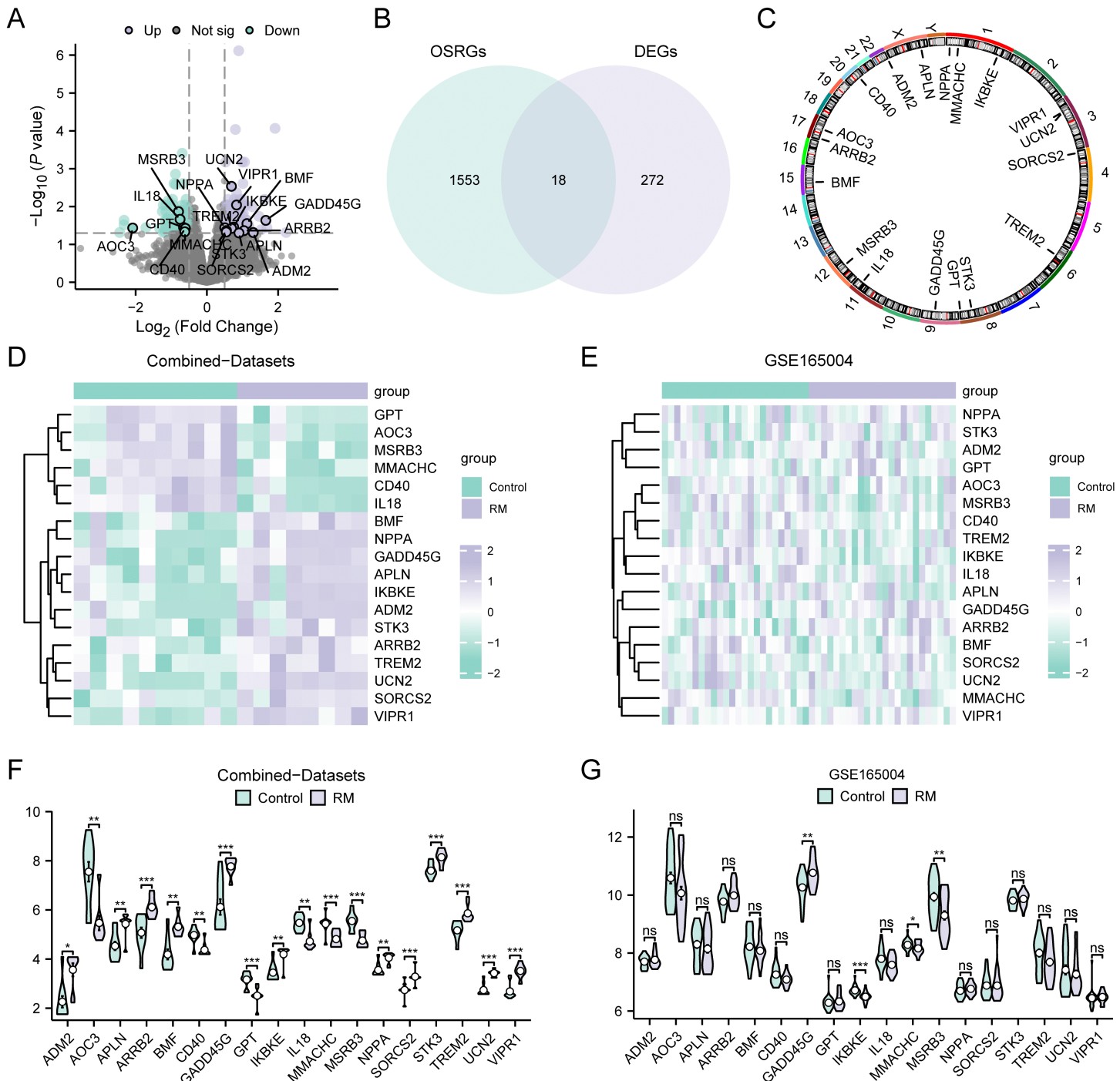

**Fig 3. Differential expression analysis of OSRDEGs. A.** Volcano plot of DEGs between the RM and control groups in the combined dataset; the OSRDEGs are highlighted as marker genes. **B.** Venn diagram illustrating the intersection of DEGs and OSRGs. **C.** Chromosomal mapping of OSR-DEGs. **D.** Heatmap of OSRDEGs between the RM and control groups in the combined dataset. **E.** Heatmap of OSRDEGs in the GSE165004 dataset between the RM and control groups. **F.** Group comparison plot of OSRDEGs expression between RM and control groups in the combined dataset. **G.** Group comparison plot of OSRDEGs expression between RM and control groups in the GSE165004 dataset. Asterisks indicate statistical significance: ns, $P \geq 0.05$ (not significant); *, $P < 0.05$; **, $P < 0.01$; ***, $P < 0.001$. DEGs, differentially expressed genes. OSRGs, oxidative stress-responsive genes. OSRDEGs, differentially expressed oxidative stress-responsive genes. RM, recurrent miscarriage.

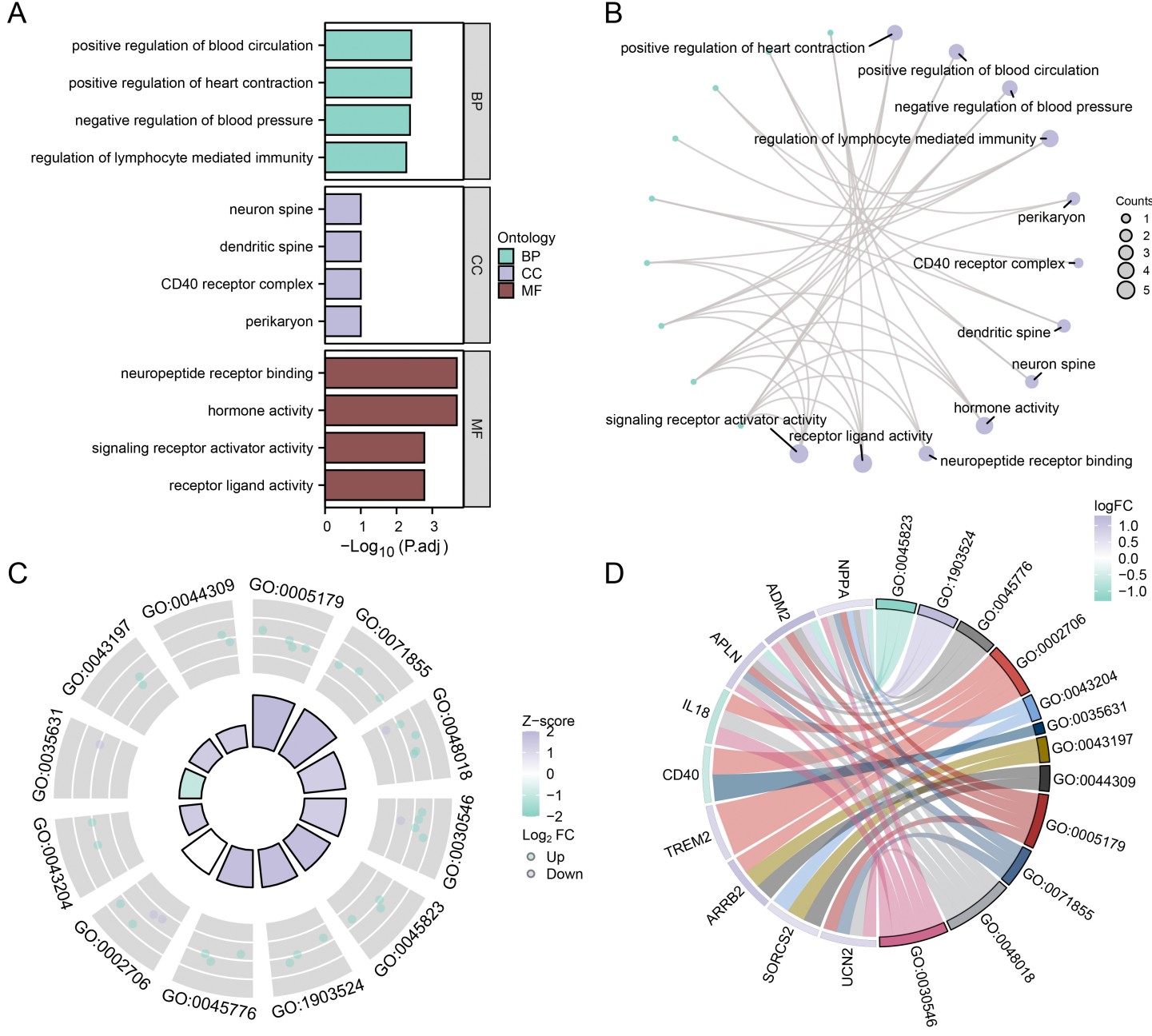

**Fig 4. GO and KEGG profiling of OSRDEGs in the RM and control groups.** **A.** Bar graph of the GO and KEGG analysis results of OSRDEGs. **B.** Network diagram of the GO and KEGG analysis results of OSRDEGs. **C.** Circle diagram of the GO and KEGG analysis of OSRDEGs combined with logFC results. **D.** Chord graph of the GO and KEGG analysis of OSRDEGs combined with logFC results. OSRDEGs, differentially expressed oxidative stress- responsive genes. GO, Gene Ontology. BP, biological process. CC, cellular component. MF, molecular function. KEGG, Kyoto Encyclopedia of Genes and Genomes. RM, recurrent miscarriage.

analyses, which are displayed as bar graphs. The network diagram in Fig 4B elucidates the intricate associations between OSRDEGs and the findings of the enrichment analyses. The lines denote the interactions between molecules, whereas the size of the nodes corresponds to the number of molecules that each entry encompasses. By employing the

benchmarks of p.adjust<0.05 and q.value<0.25 for statistical significance, the analysis revealed that the 18 OSRDEGs were enriched mainly in functions such as positive regulation of heart contraction and positive regulation of blood circulation in the context of RM. BP is associated with negative regulation of blood pressure and lymphocyte-mediated immunity. CC, perikaryon, CD40 receptor complex, dendritic spines, and neuron spines. The enriched molecular functions (MFs) that were enriched included hormone activity, neuropeptide receptor binding, receptor ligand activity, and signaling receptor activator activity (Table 3).

We subsequently performed GO and KEGG enrichment analyses of the combined logFC of these 18 OSRDEGs, and on the basis of the enrichment analysis, the corresponding z score of each gene was subsequently calculated by providing the logFC value of the OSRDEGs in the difference analysis results of the combined datasets. The results of the GO enrichment analysis of the combined logFC data are shown via a circle diagram (Fig 4C) and chord diagram (Fig 4D). Fig 4c-d shows the results of the GO enrichment analysis of the 18 OSRDEGs combined with logFC, which focused mainly on receptor ligand activity. signaling receptor activator, and other MFs. Detailed information is presented in Table 3.

We performed enrichment analyses via GO and KEGG for the logFC values consolidated from the 18 OSRDEGs. For each gene, the associated z scores were determined on the basis of the logFC values present in the differential analysis of the combined datasets. The findings from the GO and KEGG enrichment analyses, which were correlated with the logFC values, are illustrated in circular (Fig 4C) and chord diagrams (Fig 4D). These diagrams (Fig 4C-D) reveal a focus on molecular functions such as receptor ligand activity. signaling receptor activator activity, along with other MFs. Detailed information is presented in Table 3.

## GSEA profiling of OSRDEGs in the RM and control groups

To elucidate the role of gene expression in the etiology of RM and compare the RM and control groups, we employed GSEA. This analysis focused on scrutinizing the gene expression profiles and their linked biological pathways within the combined datasets. The thresholds for identifying significant enrichment were set as P.adjust<0.05 and a q value<0.25. Using these criteria, a mountain plot (Fig 5A) was constructed to visualize the results of the enrichment analysis.

**Table 3. GO and KEGG enrichment analysis results of the OSRDEGs.**

| Ontology | ID | Description | GeneRatio | BgRatio | p value | p.adjust |
|---|---|---|---|---|---|---|
| BP | GO:0045823 | positive regulation of heart contraction | 3/18 | 39/18800 | 6.59e-06 | 0.0039 |
| BP | GO:1903524 | positive regulation of blood circulation | 3/18 | 40/18800 | 7.12e-06 | 0.0039 |
| BP | GO:0045776 | negative regulation of blood pressure | 3/18 | 47/18800 | 1.16e-05 | 0.0042 |
| BP | GO:0002706 | regulation of lymphocyte mediated immunity | 4/18 | 175/18800 | 2e-05 | 0.0053 |
| BP | GO:0050731 | positive regulation of peptidyl-tyrosine phosphorylation | 4/18 | 190/18800 | 2.77e-05 | 0.0053 |
| CC | GO:0043204 | perikaryon | 2/18 | 153/19594 | 0.0085 | 0.1000 |
| CC | GO:0035631 | CD40 receptor complex | 1/18 | 11/19594 | 0.0101 | 0.1000 |
| CC | GO:0043197 | dendritic spine | 2/18 | 172/19594 | 0.0107 | 0.1000 |
| CC | GO:0044309 | neuron spine | 2/18 | 173/19594 | 0.0108 | 0.1000 |
| MF | GO:0005179 | hormone activity | 4/18 | 122/18410 | 5.23e-06 | 0.0002 |
| MF | GO:0071855 | neuropeptide receptor binding | 3/18 | 36/18410 | 5.49e-06 | 0.0002 |
| MF | GO:0048018 | receptor ligand activity | 5/18 | 489/18410 | 8.33e-05 | 0.0017 |
| MF | GO:0030546 | signaling receptor activator activity | 5/18 | 496/18410 | 8.91e-05 | 0.0017 |
| MF | GO:0001664 | G protein-coupled receptor binding | 4/18 | 288/18410 | 0.0002 | 0.0023 |

GO, Gene Ontology. BP, biological process. CC, cellular component. MF, molecular function. KEGG, Kyoto Encyclopedia of Genes and Genomes. OSRDEGs, oxidative stress-responsive differentially expressed genes.

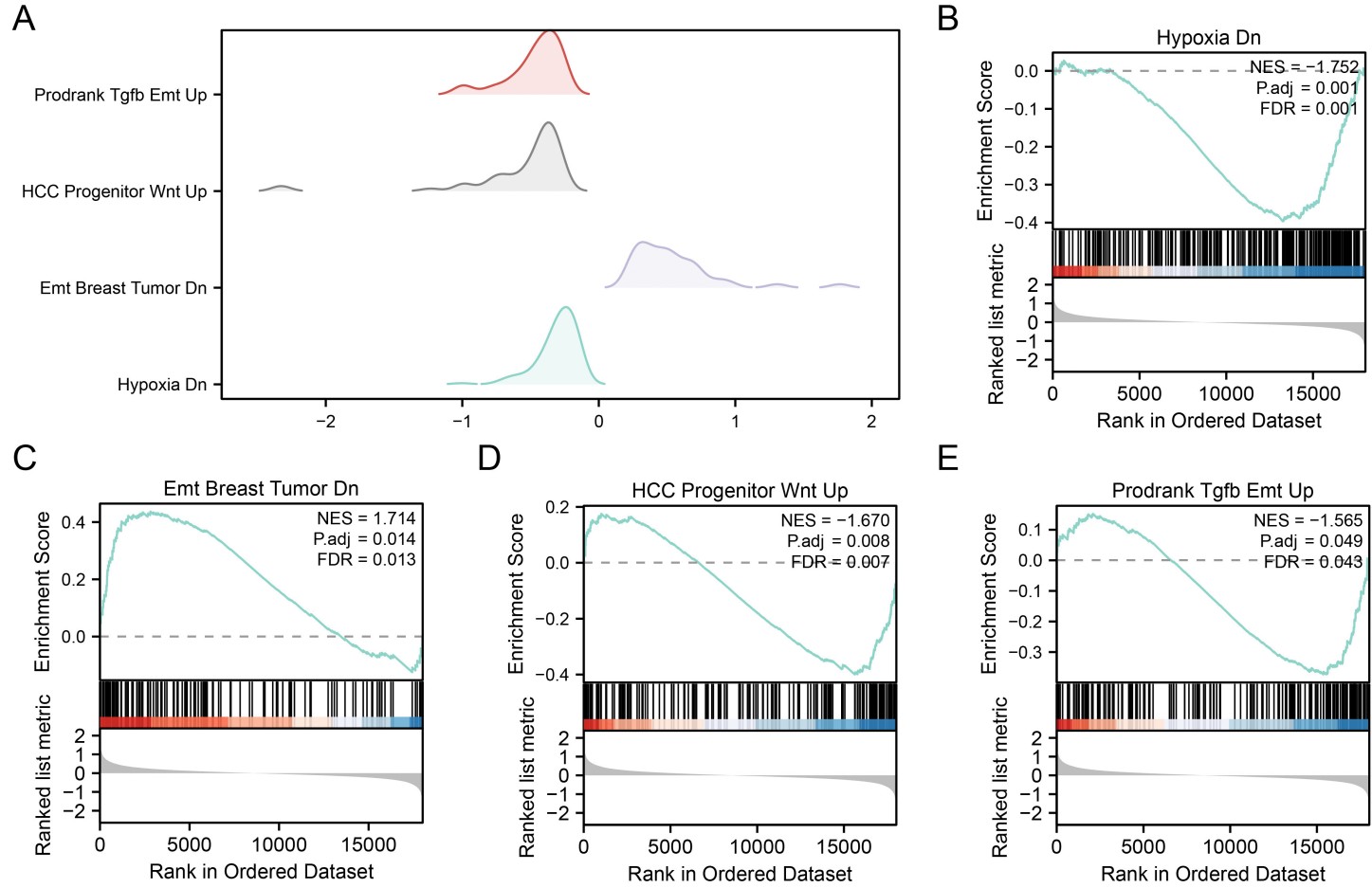

**Fig 5. GSEA profiling of OSRDEGs in the RM and control groups. A**. Mountain plot depicting the top four significantly enriched gene sets from GSEA of OSRDEGs in the combined dataset. **B-E**. OSRDEGs were significantly enriched in Hypoxia Dn (B), Emt Breast Tumor Dn (C), HCC Progenitor Wnt Up (D), and Prodrank Tgfb Emt Up (E). The significant enrichment screening criteria were p.adjust < 0.05 and q value < 0.25. GSEA, Gene Set Enrichment Analysis. OSRDEGs, oxidative stress-responsive differentially expressed genes. RM, recurrent miscarriage.

The comparative evaluation of gene expression data between the RM and control groups within combined datasets revealed considerable enrichment of genes within defined biological pathways. Specifically, the enrichment was significant for hypoxic DN (Fig 5B), epithelial–mesenchymal transition (EMT), breast tumor DN (Fig 5C), and EMT-related breast tumor DN (Fig 5C). HCC Progenitors Wnt Up (Fig 5D), Prodrinks Tgfb Emt Up (Fig 5E), and Other Pathways. Specific information is provided in Table 4.

## LASSO analysis coupled with support vector machine (SVM) modeling

Identification of 18 OSRDEGs (*ADM2*, *AOC3*, *APLN*, *ARRB2*, *BMF*, *CD40*, *GADD45G*, *GPT*, *IKBKE*, *IL18*, *MMACHC*, *MSRB3*, *NPPA*, *SORCS2*, *STK3*, *TREM2*, *UCN2*, and *VIPR1*) in the combined datasets revealed that the LASSO regression model was anchored by the genetic profiles of 18 OSRDEGs. The LASSO variable trajectory diagram (Fig 6A) and the LASSO regression model diagram (Fig 6B) were drawn for visualization. The results showed that the LASSO regression model included a total of 7 oxidative stress-responsive differentially expressed genes (OSRDEGs), which were

Table 4. Results of GSEA between the RM and control groups.

| ID | setSize | enrichmentScore | NES | p value | p.adjust | q value |
|---|---|---|---|---|---|---|
| PICCALUGA_ANGIOIMMUNOBLASTIC_LYMPHOMA_UP | 194 | −0.5973811 | −2.533554 | 1e-10 | 5.42e-08 | 4.8e-08 |
| SMID_BREAST_CANCER_NORMAL_LIKE_UP | 434 | −0.5337110 | −2.465749 | 1e-10 | 5.42e-08 | 4.8e-08 |
| LINDGREN_BLADDER_CANCER_CLUSTER_2B | 361 | −0.5184478 | −2.357413 | 1e-10 | 5.42e-08 | 4.8e-08 |
| BOQUEST_STEM_CELL_UP | 251 | −0.5374932 | −2.339802 | 1e-10 | 5.42e-08 | 4.8e-08 |
| LIU_PROSTATE_CANCER_DN | 459 | −0.4816837 | −2.233316 | 1e-10 | 5.42e-08 | 4.8e-08 |
| ONDER_CDH1_TARGETS_2_UP | 243 | −0.5101658 | −2.225963 | 1e-10 | 5.42e-08 | 4.8e-08 |
| SCHUETZ_BREAST_CANCER_DUCTAL_INVASIVE_UP | 333 | −0.4837525 | −2.174587 | 1e-10 | 5.42e-08 | 4.8e-08 |
| LIM_MAMMARY_STEM_CELL_UP | 445 | −0.4631274 | −2.143688 | 1e-10 | 5.42e-08 | 4.8e-08 |
| CHARAFE_BREAST_CANCER_LUMINAL_VS_MESENCHYMAL_DN | 434 | −0.4541212 | −2.098043 | 1e-10 | 5.42e-08 | 4.8e-08 |
| RODWELL_AGING_KIDNEY_UP | 446 | −0.4215364 | −1.950038 | 1e-10 | 5.42e-08 | 4.8e-08 |
| HOLLERN_EMT_BREAST_TUMOR_DN | 115 | 0.436697717 | 1.71376536 | 0.000301871 | 0.014486896 | 0.012831645 |
| FOROUTAN_PRODRANK_TGFB_EMT_UP | 183 | −0.372932787 | −1.564882624 | 0.001700984 | 0.048504901 | 0.042962803 |
| MEBARKI_HCC_PROGENITOR_WNT_UP | 175 | −0.399429231 | −1.670371179 | 0.000109243 | 0.007891708 | 0.006990013 |
| MANALO_HYPOXIA_DN | 266 | −0.396374014 | −1.751908414 | 8.4664E-06 | 0.001176178 | 0.00104179 |

GSEA: Gene Set Enrichment Analysis. RM, recurrent miscarriage.

*ARRB2*, *BMF*, *IL18*, *SORCS2*, *STK3*, *UCN2*, and *VIPR1*. To further verify the value of the RM diagnostic model, a forest map was drawn based on the seven OSRDEGs (Fig 6C).

On the basis of 18 OSRDEGs (*ADM2*, *AOC3*, *APLN*, *ARRB2*, *BMF*, *CD40*, *GADD45G*, *GPT*, *IKBKE*, *IL18*, *MMACHC*, *MSRB3*, *NPPA*, *SORCS2*, *STK3*, *TREM2*, *UCN2*, and *VIPR1*), an SVM methodology was implemented to develop the SVM model, and the collection of genes with the lowest error rates (Fig 6D), and the highest accuracy rates (Fig 6E) were identified. The results indicated that the SVM model achieved peak precision with a gene count of 10, and the 10 genes were *STK3*, *VIPR1*, *TREM2*, *SORCS2*, *MSRB3*, *UCN2*, *ARRB2*, *GADD45G*, *BMF*, and *GPT*.

Next, to obtain the common OSRDEGs defined as hub genes, we intersected the OSRDEGs from the LASSO regression model with those from the random forest model (RF) and constructed a Venn diagram (Fig 6F) to identify a total of 6 hub genes, which were as follows: *ARRB2*, *BMF*, *SORCS2*, *STK3*, *UCN2*, and *VIPR1*.

Finally, to evaluate the comparative significance of the hub genes via the GOSemSim package, we performed an analysis focusing on the functional resemblance among these central genes and displayed the results via a functional similarity analysis box diagram (Fig 6G). The results showed that *VIPR1* and *BMF* were relatively more important for function.

## Construction and diagnostic performance of the hub gene diagnostic model

To ascertain the diagnostic potential of the six hub genes (*ARRB2*, *BMF*, *SORCS2*, *STK3*, *UCN2*, and *VIPR1*) in the combined datasets, we performed logistic regression analysis to develop a forecasting model. This model was based on these six hub genes, and RM samples were stratified into high-risk (High) and low-risk (Low) groups via the median forecast derived from the model's analysis. The logistic regression model included six hub genes. To further validate the diagnostic value of this model, we constructed a forest map (Fig 7A) and a nomogram (Fig 7B) to illustrate the contribution of these genes to the model. The logistic regression model revealed that the predictive efficacy of *STK3* gene expression markedly surpassed that of other variables. The formula for ascertaining the forecasted worth is as follows:

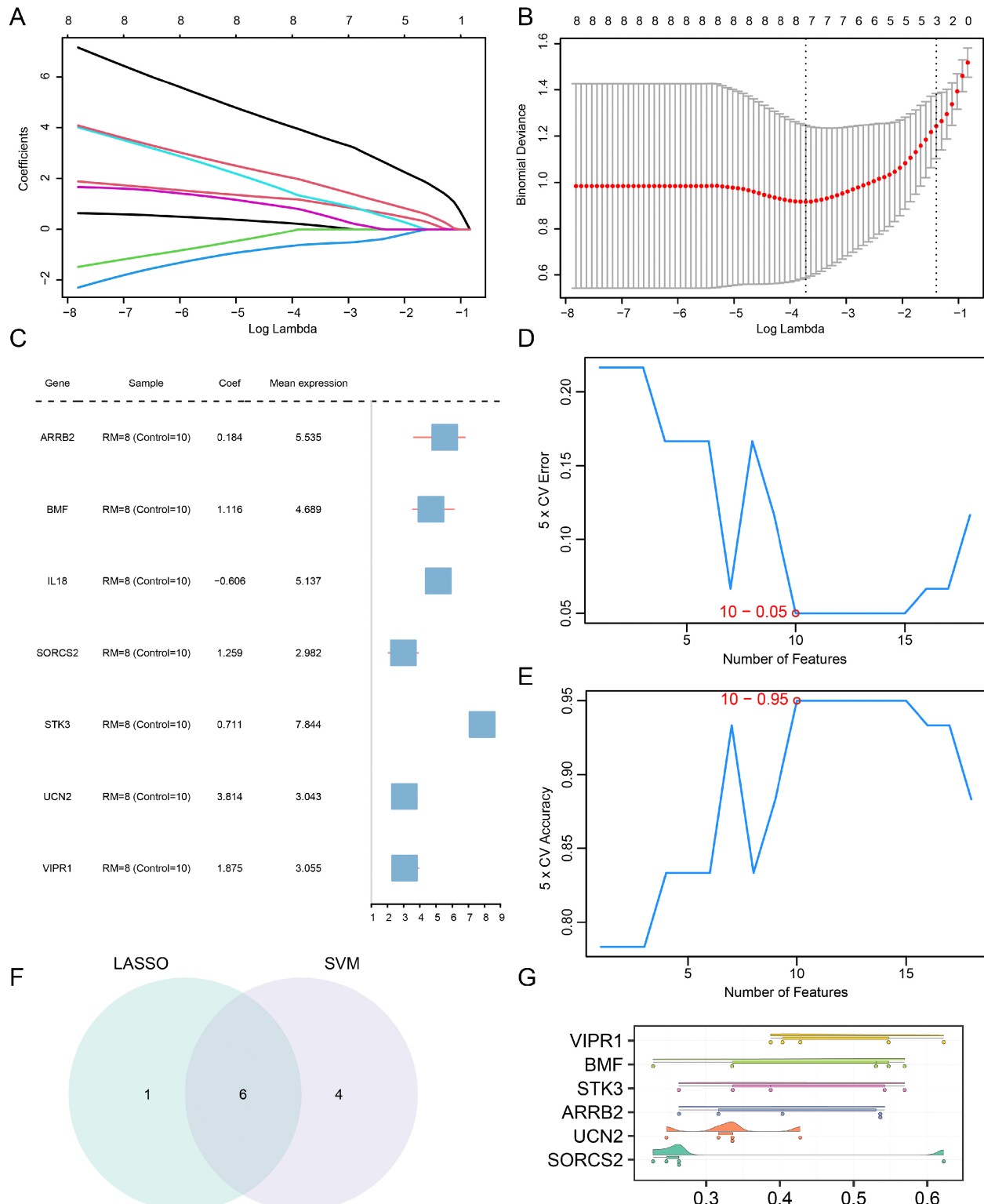

**Fig 6. LASSO analysis coupled with SVM modeling. A.** Variable trajectory plot of the LASSO regression model. **B.** Diagnostic model plot of the LASSO regression model. **C.** Forest plot of the LASSO regression model. **D-E.** Plots of the cross-validation error rate (D) and accuracy (E) against the number of genes in the SVM model. F. OSRDEGs selected by the LASSO regression model and SVM Venn diagram. G. Box plot for functional similarity analysis of hub genes. OSRDEGs, differentially expressed oxidative stress-responsive genes. LASSO, least absolute shrinkage and selection operator. SVM, support vector machine.

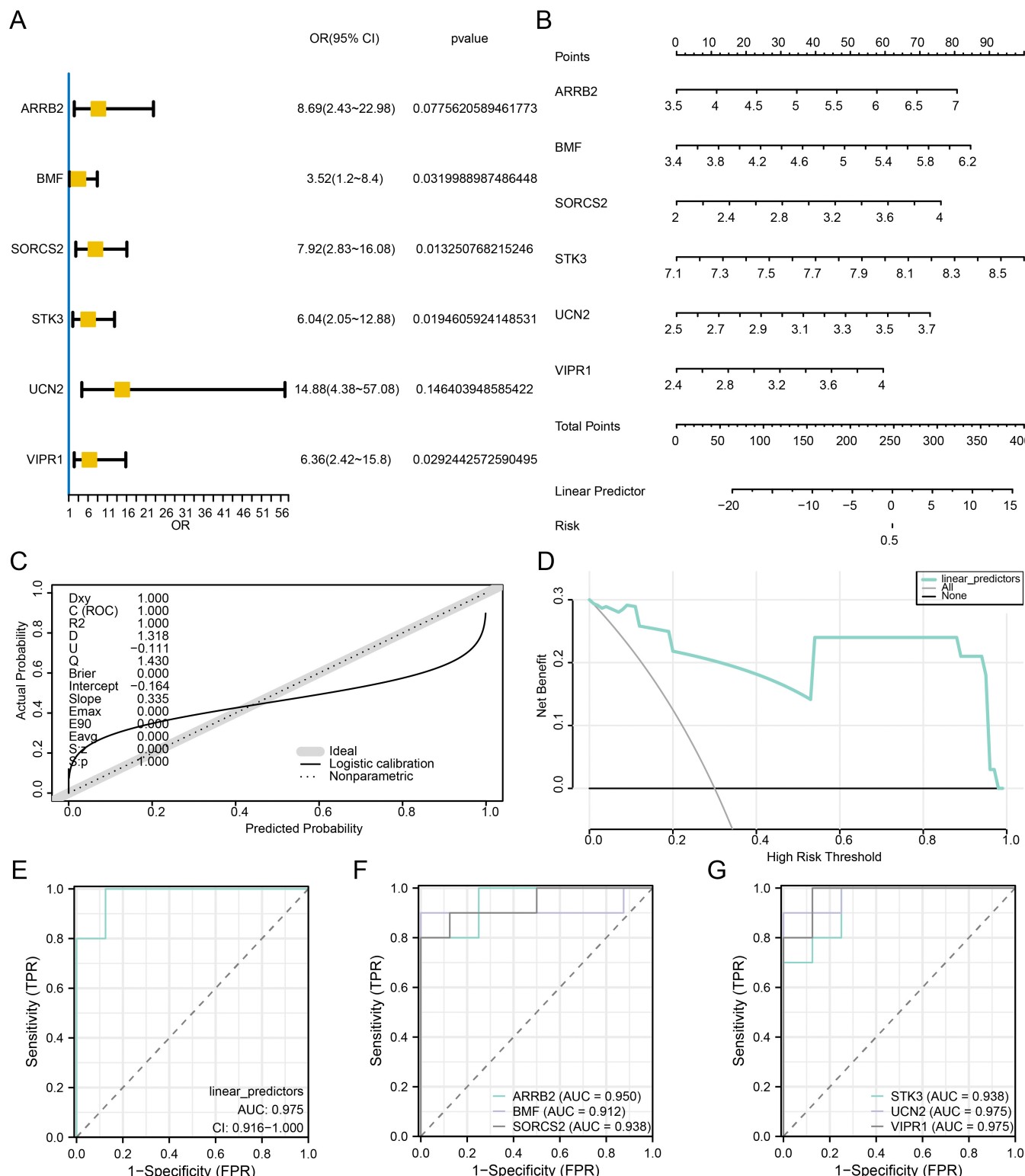

**Fig 7. Construction and diagnostic performance of the hub gene diagnostic model. A-B.** Forest plot (A) and nomogram (B) of the 6 hub genes in the logistic regression model. **C-D**. Calibration plot (C) and DCA plot (D) of the logistic regression model. **E.** ROC curves of the logistic model prediction values in the combined datasets. **F.** ROC curves of *ARRB2*, *BMF*, and *SORCS2* expression in the combined datasets. G. ROC curves of *STK3*, *UCN2*, and *VIPR1* gene expression in the combined dataset. AUC, area under the curve. DCA, decision curve analysis. ROC, receiver operating characteristic curve.

$$\text{linear predictors} = \sum_i Coefficient\ (gene_i) * mRNA\ Expression\ (gene_i)$$

Finally, to assess the correctness and discriminatory power of the logistic regression model, we constructed a calibration curve via a calibration analysis. The predictive performance of the model against the actual outcomes was assessed by comparing the optimal theoretical probabilities (solid lines) with the probabilities predicted by the model (dashed lines) across the various scenarios depicted in the figure (Fig 7C). The clinical effectiveness of the logistic regression models was assessed via decision curve analysis (DCA), and the findings are displayed in Fig 7D. In the DCA figure, if the model's curve surpasses both the all-positive and all-negative lines, the broader the interval is, the greater the net benefits, and the more a model performs. On the basis of these findings (Fig 7C-D), the model constructed in this study has high accuracy in the diagnosis of recurrent spontaneous abortion.

For the purpose of additional validation of the logistic model and the central gene diagnostic model, receiver operating characteristic (ROC) analysis was performed, incorporating the forecasted probabilities of the logistic model, the gene expression levels of the six hub genes, and the group (RM/Control) details from the combined datasets. The outcomes revealed that the forecasted efficacy of the logistic model (AUC = 0.975, Fig 7E) was highly accurate in diagnosing the RM/control groups. The diagram in Fig 7F indicates that the levels of *ARRB2* (AUC = 0.960), *BMF* (AUC = 0.912), and *SORCS2* (AUC = 0.938) had increased diagnostic accuracy in identifying the RM/control groups. The diagram in Fig 7G shows that the expression levels of *STK3* (AUC = 0.936), *UCN2* (AUC = 0.975), and *VIPR1* (AUC = 0.975) exhibited high diagnostic accuracy in distinguishing between the RM and control groups.

## Hub gene correlation analysis

By utilizing the gene expression matrix for the six hub genes (*ARRB2*, *BMF*, *SORCS2*, *STK3*, *UCN2*, and *VIPR1*) from both the combined datasets and the GSE165004 dataset, we initially employed Spearman's rank correlation method to assess the relationships between the expression levels of the six hub genes, subsequently presenting the outcomes in a correlation heatmap (Fig 8A-B).

The results indicated that the expression patterns of the six hub genes identified in the combined datasets (Fig 8A) and GSE165004 datasets (Fig 8B) were significantly correlated with the expression levels of the majority of the other hub genes. According to the results of the correlation heatmap in Fig 8A and Fig 8B, we selected the gene combinations with the same significant correlation in the combined datasets and GSE165004 dataset and plotted the correlation scatter plot (Fig 8B). C-L. In the combined datasets, there was a slight positive correlation between *BMF* and *ARRB2* (R = 0.474, P = 0.049; Fig 8C) and an intermediate positive correlation between *SORCS2* and *ARRB2* (R = 0.701, P = 0.002; Fig 8D). There was an intermediate positive correlation between *SORCS2* and *BMF* (R = 0.583, P = 0.013; Fig 8E) and between *UCN2* and *ARRB2* (R = 0.748, P < 0.001; Fig 8F). An intermediate positive correlation was observed between *UCN2* and *SORCS2* expression (R = 0.562, P = 0.017; Fig 8G). In dataset GSE165004, there was a slight positive correlation between *BMF* and *ARRB2* (R = 0.360, P = 0.012; Fig 8H), and a slight positive correlation was observed between *SORCS2* and *ARRB2* (R = 0.373, P = 0.009; Fig 8I). There was an intermediate positive correlation between *SORCS2* and *BMF* (R = 0.637, P < 0.001; Fig 8J) and between *UCN2* and *ARRB2* (R = 0.629, P < 0.001; Fig 8K). An intermediate positive correlation was observed between *UCN2* and *SORCS2* (R = 0.646, P < 0.001; Fig 8L).

## Combined datasets GSEA profiling of OSRDEGs in the high and low groups

To determine the link between gene expression patterns and predictive values in the high- and low-BP groups stratified by the logistic regression model, we performed GSEA on the combined datasets to investigate the interplay between gene expression and associated BPs, CCs, and MFs. The thresholds for identifying significant enrichment were set as P.adjust < 0.05 and a q value < 0.25. A mountain plot (Fig 9A) was constructed to visualize the results of the enrichment

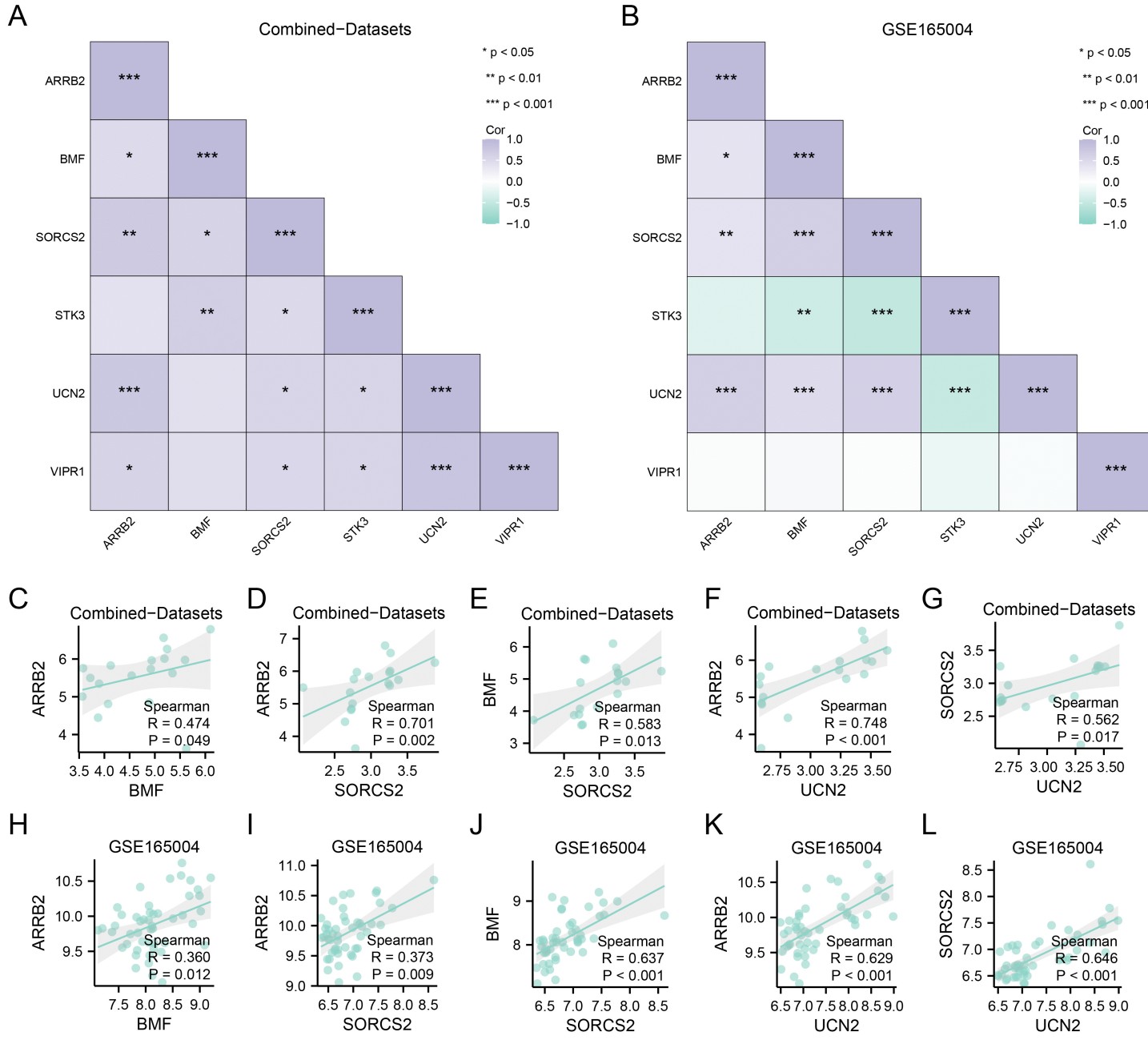

**Fig 8. Correlation analysis of the hub genes. A-B**. Correlation heatmap of the hub genes based on Spearman's rank correlation analysis in the combined dataset (A) and GSE165004(B) dataset. **C–G**. Scatter plots with correlations in the combined dataset: *BMF* and *ARRB2* (C), *SORCS2* and *ARRB2* (D), *SORCS2* and *BMF* (E), *UCN2* and *ARRB2* (F), and *UCN2* and *SORCS2* (G). **H-L**. Scatter plots with correlations in the GSE165004 dataset: *BMF* and *ARRB2* (H), *SORCS2* and *ARRB2* (I), *SORCS2* and *BMF* (J), *UCN2* and *ARRB2* (K), and *UCN2* and *SORCS2* (L). Asterisks in the correlation heatmap indicate statistical significance: ns, P ≥ 0.05 (not significant); *, P < 0.05; **, P < 0.01; ***, P < 0.001.

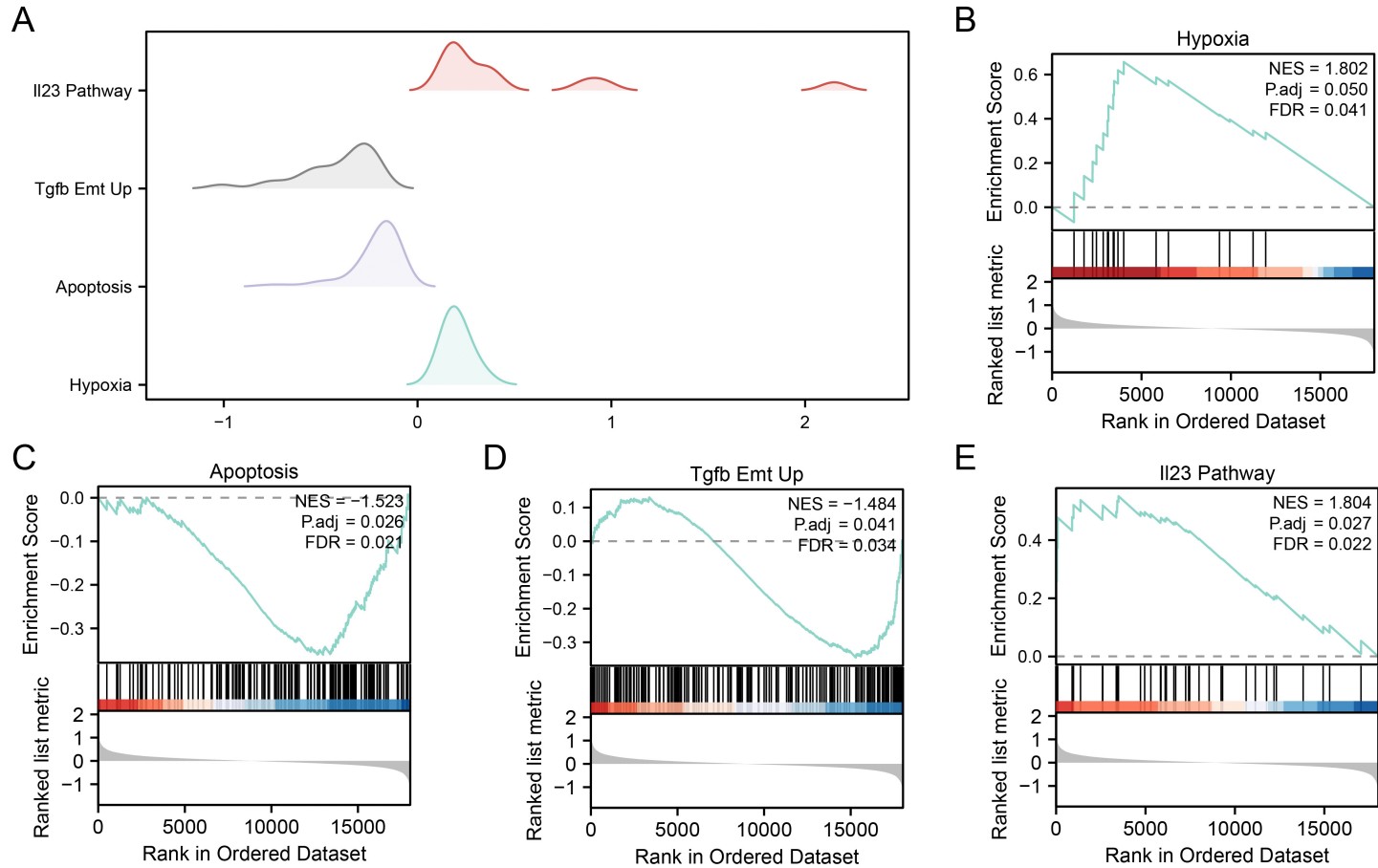

**Fig 9. Combined dataset GSEA profiling of OSRDEGs in the high and low groups. A.** Mountain plot displaying the top four significantly enriched gene sets from GSEA between the high and low groups in the combined dataset. **B-E**. Genes in the high and low groups in the combined dataset were significantly enriched in the hypoxia (B), apoptosis (C), Tgfb Emt Up (D), and Il23 pathways (E). GSEA, Gene Set Enrichment Analysis. The significant enrichment screening criteria: p.adjust < 0.05 and FDR value (q value) < 0.25.

analysis. The results revealed that the genes separating the high and low groups in the combined datasets were substantially overrepresented in pathways related to hypoxia (Fig 9B), apoptosis (Fig 9C), Tgfb Emt Up (Fig 9D), the Il23 pathway (Fig 9E), and other relevant pathways. Further details are presented in Table 5.

## Development of mRNA–miRNA and mRNA–TF interaction networks

We harnessed mRNA–miRNA information from the miRDB database to predict miRNAs that engage with six hub genes (*ARRB2*, *BMF*, *SORCS2*, *STK3*, *UCN2*, and *VIPR1*). The interaction network of mRNAs and miRNAs was subsequently constructed via Cytoscape software (Fig 10A). The mRNA–miRNA interaction network is diagrammed, with purple ovals indicating mRNAs and green ovals indicating miRNAs. The mRNA–miRNA interaction network included five hub genes (*ARRB2*, *BMF*, *SORCS2*, *STK3*, and *UCN2*) and 41 miRNA molecules, resulting in 41 mRNA–miRNA interaction associations. *STK3*, a hub gene within the mRNA–miRNA interaction network, exhibited the highest level of interaction with miRNAs and was associated with 17 TFs. The specific information is presented in Table 6.

We explored the CHIPBase and HTF target databases to identify the TFs associated with binding to hub genes (*ARRB2*, *BMF*, *SORCS2*, *STK3*, *UCN2*, and *VIPR1*). We retrieved interaction data from both databases, calculated their

**Table 5. Results of GSEA gene set enrichment analysis between the high and low groups.**

| ID | set-Size | enrichment-Score | NES | P value | p.adjust | q value |
|---|---|---|---|---|---|---|
| ROSTY_CERVICAL_CANCER_PROLIFERATION_CLUSTER | 134 | −0.7556770 | −3.109515 | 1e-10 | 7.63e-09 | 6.25e-09 |
| SOTIRIOU_BREAST_CANCER_GRADE_1_VS_3_UP | 143 | −0.7330559 | −3.040925 | 1e-10 | 7.63e-09 | 6.25e-09 |
| FLORIO_NEOCORTEX_BASAL_RADIAL_GLIA_DN | 172 | −0.7008142 | −2.970941 | 1e-10 | 7.63e-09 | 6.25e-09 |
| GRAHAM_NORMAL_QUIESCENT_VS_NORMAL_DIVIDING_DN | 86 | −0.7566878 | −2.895701 | 1e-10 | 7.63e-09 | 6.25e-09 |
| LEE_EARLY_T_LYMPHOCYTE_UP | 101 | −0.7326222 | −2.857588 | 1e-10 | 7.63e-09 | 6.25e-09 |
| WHITEFORD_PEDIATRIC_CANCER_MARKERS | 109 | −0.7203380 | −2.848725 | 1e-10 | 7.63e-09 | 6.25e-09 |
| CROONQUIST_IL6_DEPRIVATION_DN | 92 | −0.7428382 | −2.841677 | 1e-10 | 7.63e-09 | 6.25e-09 |
| KOBAYASHI_EGFR_SIGNALING_24HR_DN | 243 | −0.6424261 | −2.835825 | 1e-10 | 7.63e-09 | 6.25e-09 |
| KONG_E2F3_TARGETS | 91 | −0.7375312 | −2.821733 | 1e-10 | 7.63e-09 | 6.25e-09 |
| ZHAN_MULTIPLE_MYELOMA_PR_UP | 43 | −0.8428701 | −2.811281 | 1e-10 | 7.63e-09 | 6.25e-09 |
| PID_IL23_PATHWAY | 37 | 0.551012346 | 1.804274069 | 0.002171066 | 0.027291963 | 0.022338995 |
| KIM_HYPOXIA | 17 | 0.656862496 | 1.802398332 | 0.005056878 | 0.049724434 | 0.040700402 |
| FOROUTAN_TGFB_EMT_UP | 190 | −0.345526281 | −1.483649602 | 0.003948922 | 0.041144733 | 0.033677753 |
| REACTOME_APOPTOSIS | 163 | −0.360568681 | −1.523275047 | 0.00206213 | 0.026226805 | 0.021467142 |

GSEA, gene set enrichment analysis.

intersections, and identified six hub genes (*ARRB2, BMF, SORCS2, STK3, UCN2, and VIPR1*) along with 94 transcription factors. A total of 201 mRNA–TF interaction associations were established and graphically represented via Cytoscape. (Fig 10B). The mRNA–TF interaction network is diagrammed, with purple ovals indicating mRNAs and orange ovals indicating miRNAs. *BMF*, a hub gene within the mRNA–TF interaction network, exhibited the highest level of interaction with TFs and was associated with 56 TFs. The specific information is presented in Table 7.

### Analysis of immune system infiltration across the high and low groups in the combined datasets (CIBERSORTx)

Immune cells play crucial roles in the onset and progression of depression. We evaluated the status of 22 immune cell distributions via CIBERSORT analysis and evaluated the prevalence of diverse immune cell populations via the Wilcoxon test. The level of immune cell distribution in the combined datasets was evaluated **across** the high and low groups (Fig 11A). The results revealed that 22 types of immune cells (B.cells.naive, B.cells.memory, Plasma.cells, T.cells.CD8, T.cells.CD4.naive, T.cells.CD4.memory.resting, T.cells.CD4.memory.activated, T.cells.follicular.helper, T.cells.regulatory.Tregs., T.cells.gamma.delta, NK.cells.resting, NK.cells.activated, Monocytes, Macrophages. M0, Macrophages. M1, Macrophages. M2, Dendritic.cells.resting, Dendritic.cells.activated, Mast.cells.resting, Mast.cells.activated, Eosinophils, Neutrophils) were differentially abundant across the high and low groups. We created a heatmap (Fig 11B) illustrating the associations between the abundance of infiltrating immune cells and six hub genes (*ARRB2*, *BMF*, *SORCS2*, *STK3*, *UCN2*, and *VIPR1*). Plasma cells, T.cells.CD8, monocytes, and neutrophils were significantly correlated with *ARRB2* and *UCN2*, and a correlation scatter plot was created (Fig 11C-F), which revealed an intermediate negative correlation between *ARRB2* and plasma cells (R=−0.714, P=0.047; Fig 11C). There was an intermediate positive correlation between *the UCN2* and T. cells.CD8 (R=0.791, P=0.019; Fig 11D). There was a pronounced negative correlation between *UCN2* and monocyte count (R=−0.810, P=0.022; Fig 11E). There was a pronounced negative correlation between *ARRB2* and the neutrophil count (R=−0.976, P<0.001; Fig 11F).

### Immunohistochemical analysis of hub protein expression

IHC analysis revealed significantly higher expression levels of six proteins (*ARRB2, SORCS2, VIPR1, UCN2, STK3* and *BMF*) in the endometrial tissues of the RM group than in those of the control group, as detailed in Fig 12.

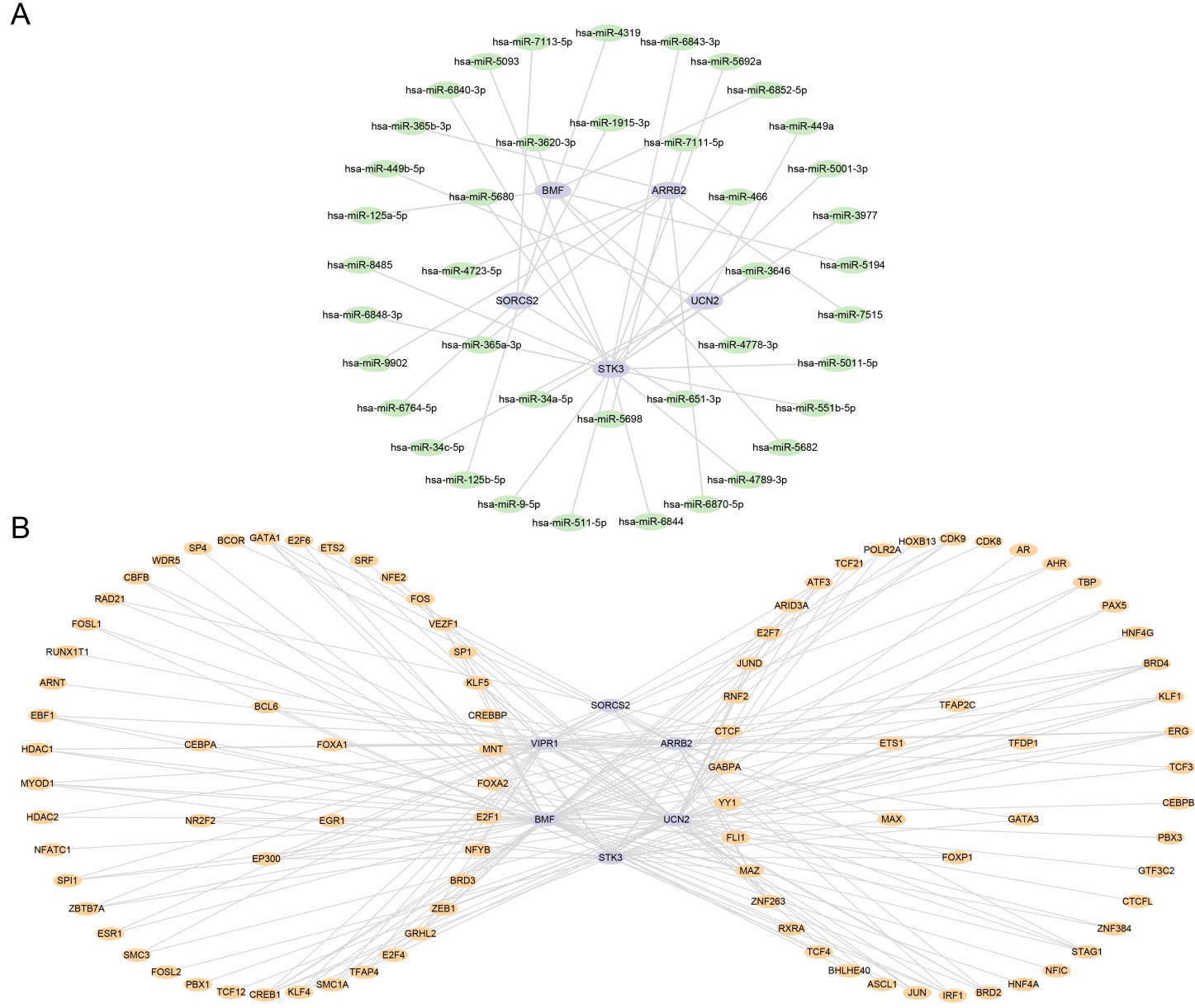

**Fig 10. Construction of mRNA-miRNA and mRNA-TF interaction networks for the hub genes. A.** The mRNA-miRNA interaction network of the hub genes and miRNAs. Nodes represent molecules (purple ovals: hub gene mRNAs; green ovals: miRNAs). **B.** The mRNA-TF interaction network. Nodes represent molecules (purple ovals: hub gene mRNAs; orange ovals: TFs). TF, transcription factor.

## Discussion

RM, which is characterized by the loss of two or more pregnancies, is a troubling reproductive disorder that affects approximately 1–2% of women of reproductive age [2]. RM profoundly influences the emotional and psychological states of affected individuals [34]. Despite its prevalence and impact, the underlying causes of RM remain poorly understood, necessitating further research on its pathophysiology and potential therapeutic targets.

**Table 6. List of miRNA-related hub genes.**

| mRNA | miRNA |
| --- | --- |
| ARRB2 | hsa-miR-365a-3p |
| ARRB2 | hsa-miR-365b-3p |
| ARRB2 | hsa-miR-9902 |
| ARRB2 | hsa-miR-5698 |
| ARRB2 | hsa-miR-7111-5p |
| ARRB2 | hsa-miR-6870-5p |
| ARRB2 | hsa-miR-7515 |
| ARRB2 | hsa-miR-4723-5p |
| BMF | hsa-miR-4319 |
| BMF | hsa-miR-125a-5p |
| BMF | hsa-miR-125b-5p |
| BMF | hsa-miR-4778-3p |
| BMF | hsa-miR-3620-3p |
| BMF | hsa-miR-5682 |
| BMF | hsa-miR-5194 |
| BMF | hsa-miR-6852-5p |
| SORCS2 | hsa-miR-7113-5p |
| SORCS2 | hsa-miR-6764-5p |
| SORCS2 | hsa-miR-651-3p |
| SORCS2 | hsa-miR-1915-3p |
| STK3 | hsa-miR-4789-3p |
| STK3 | hsa-miR-3977 |
| STK3 | hsa-miR-5692a |
| STK3 | hsa-miR-5093 |
| STK3 | hsa-miR-8485 |
| STK3 | hsa-miR-9-5p |
| STK3 | hsa-miR-3646 |
| STK3 | hsa-miR-5680 |
| STK3 | hsa-miR-551b-5p |
| STK3 | hsa-miR-5001-3p |
| STK3 | hsa-miR-6843-3p |
| STK3 | hsa-miR-6840-3p |
| STK3 | hsa-miR-6848-3p |
| STK3 | hsa-miR-511-5p |
| STK3 | hsa-miR-466 |
| STK3 | hsa-miR-6844 |
| STK3 | hsa-miR-5011-5p |
| UCN2 | hsa-miR-449a |
| UCN2 | hsa-miR-34a-5p |
| UCN2 | hsa-miR-449b-5p |
| UCN2 | hsa-miR-34c-5p |

**Table 7. List of TF-related hub genes.**

| mRNA | TF |
| --- | --- |
| ARRB2 | E2F1 |
| ARRB2 | E2F6 |
| ARRB2 | EBF1 |
| ARRB2 | EP300 |
| ARRB2 | ERG |
| ARRB2 | ETS1 |
| ARRB2 | FLI1 |
| ARRB2 | FOXA1 |
| ARRB2 | FOXA2 |
| ARRB2 | GATA1 |
| ARRB2 | HDAC1 |
| ARRB2 | IRF1 |
| ARRB2 | KLF1 |
| ARRB2 | MAX |
| ARRB2 | MAZ |
| ARRB2 | BCL6 |
| ARRB2 | MNT |
| ARRB2 | BCOR |
| ARRB2 | MYOD1 |
| ARRB2 | BRD2 |
| ARRB2 | BRD4 |
| ARRB2 | CDK9 |
| ARRB2 | ZNF263 |
| ARRB2 | CREB1 |
| ARRB2 | CREBBP |
| BMF | HDAC1 |
| BMF | HDAC2 |
| BMF | HNF4A |
| BMF | HNF4G |
| BMF | HOXB13 |
| BMF | IRF1 |
| BMF | KLF1 |
| BMF | KLF4 |
| BMF | KLF5 |
| BMF | MAZ |
| BMF | MYOD1 |
| BMF | NFATC1 |
| BMF | NFIC |
| BMF | PAX5 |
| BMF | POLR2A |
| BMF | RXRA |
| BMF | SMC1A |
| BMF | SP1 |
| BMF | SP4 |
| BMF | SPI1 |
| BMF | STAG1 |

*(Continued)*

**Table 7.** (Continued)

| mRNA | TF |
|------|-----|
| BMF | TBP |
| BMF | TCF21 |
| BMF | TCF4 |
| BMF | TFAP4 |
| BMF | VEZF1 |
| BMF | WDR5 |
| BMF | ZBTB7A |
| BMF | ZNF384 |
| BMF | AHR |
| BMF | ATF3 |
| BMF | BHLHE40 |
| BMF | BRD2 |
| BMF | BRD3 |
| BMF | BRD4 |
| BMF | CBFB |
| BMF | CDK9 |
| BMF | CEBPA |
| BMF | CEBPB |
| BMF | CREB1 |
| BMF | CREBBP |
| BMF | CTCF |
| BMF | E2F1 |
| BMF | E2F4 |
| BMF | E2F6 |
| BMF | EBF1 |
| BMF | EP300 |
| BMF | ERG |
| BMF | ESR1 |
| BMF | FLI1 |
| BMF | FOXA1 |
| BMF | FOXA2 |
| BMF | FOXP1 |
| BMF | GABPA |
| BMF | GATA1 |
| BMF | GRHL2 |
| SORCS2 | CTCF |
| SORCS2 | RAD21 |
| SORCS2 | SMC3 |
| SORCS2 | STAG1 |
| SORCS2 | ZBTB7A |
| SORCS2 | ZNF263 |
| STK3 | ARID3A |
| STK3 | EP300 |
| STK3 | ERG |
| STK3 | ETS1 |

*(Continued)*

**Table 7.** (Continued)

| mRNA | TF |
|---|---|
| *STK3* | FLI1 |
| *STK3* | FOS |
| *STK3* | FOSL1 |
| *STK3* | FOSL2 |
| *STK3* | FOXP1 |
| *STK3* | GABPA |
| *STK3* | GATA1 |
| *STK3* | GATA3 |
| *STK3* | ATF3 |
| *STK3* | IRF1 |
| *STK3* | JUN |
| *STK3* | KLF1 |
| *STK3* | KLF5 |
| *STK3* | MAX |
| *STK3* | MAZ |
| *STK3* | MYOD1 |
| *STK3* | BCL6 |
| *STK3* | NR2F2 |
| *STK3* | BRD4 |
| *STK3* | AR |
| *STK3* | CREB1 |
| *UCN2* | ASCL1 |
| *UCN2* | ATF3 |
| *UCN2* | BHLHE40 |
| *UCN2* | BRD2 |
| *UCN2* | BRD4 |
| *UCN2* | CBFB |
| *UCN2* | CREB1 |
| *UCN2* | CREBBP |
| *UCN2* | CTCF |
| *UCN2* | CTCFL |
| *UCN2* | E2F1 |
| *UCN2* | E2F4 |
| *UCN2* | E2F7 |
| *UCN2* | EBF1 |
| *UCN2* | EGR1 |
| *UCN2* | ERG |
| *UCN2* | ETS1 |
| *UCN2* | FLI1 |
| *UCN2* | FOSL1 |
| *UCN2* | FOXA2 |
| *UCN2* | GABPA |
| *UCN2* | GATA1 |
| *UCN2* | GTF3C2 |
| *UCN2* | HDAC1 |

*(Continued)*

| mRNA | TF |
|------|-----|
| UCN2 | JUN |
| UCN2 | JUND |
| UCN2 | KLF1 |
| UCN2 | KLF4 |
| UCN2 | MAZ |
| UCN2 | MYOD1 |
| UCN2 | NFE2 |
| UCN2 | NR2F2 |
| UCN2 | PAX5 |
| UCN2 | PBX1 |
| UCN2 | PBX3 |
| UCN2 | POLR2A |
| UCN2 | RAD21 |
| UCN2 | RNF2 |
| UCN2 | SMC1A |
| UCN2 | SMC3 |
| UCN2 | SP1 |
| UCN2 | SPI1 |
| UCN2 | SRF |
| UCN2 | STAG1 |
| UCN2 | TBP |
| UCN2 | TCF12 |
| UCN2 | TCF21 |
| UCN2 | TCF3 |
| UCN2 | TFDP1 |
| UCN2 | YY1 |
| UCN2 | ZBTB7A |
| UCN2 | ZEB1 |
| UCN2 | ZNF384 |
| VIPR1 | AHR |
| VIPR1 | ARNT |
| VIPR1 | ATF3 |
| VIPR1 | BRD2 |
| VIPR1 | BRD4 |
| VIPR1 | CDK8 |
| VIPR1 | CDK9 |
| VIPR1 | CREB1 |
| VIPR1 | CTCF |
| VIPR1 | EP300 |
| VIPR1 | ERG |
| VIPR1 | ESR1 |
| VIPR1 | ETS2 |
| VIPR1 | FOXA1 |
| VIPR1 | FOXA2 |
| VIPR1 | FOXP1 |

*(Continued)*

**Table 7.** (Continued)

| mRNA | TF |
|------|-----|
| *VIPR1* | GABPA |
| *VIPR1* | GATA3 |
| *VIPR1* | GRHL2 |
| *VIPR1* | HDAC1 |
| *VIPR1* | HDAC2 |
| *VIPR1* | IRF1 |
| *VIPR1* | JUN |
| *VIPR1* | KLF4 |
| *VIPR1* | KLF5 |
| *VIPR1* | MAZ |
| *VIPR1* | NFYB |
| *VIPR1* | NR2F2 |
| *VIPR1* | RUNX1T1 |
| *VIPR1* | SMC1A |
| *VIPR1* | SP1 |
| *VIPR1* | SPI1 |
| *VIPR1* | STAG1 |
| *VIPR1* | TCF3 |
| *VIPR1* | TFAP2C |
| *VIPR1* | TFAP4 |

TF, Transcription factor.

Exploring the differential expression of OSRGs in RM offers a promising avenue for understanding the pathophysiology of this disease. OS is associated with placental dysfunction and pregnancy complications [5], leading us to hypothesize that altered OSRG expression contributes to RM development. This study aimed to uncover the molecular basis of RM by examining the differential expression and functional roles of these genes and potentially identifying new biomarkers for diagnosis and treatment. These findings may ultimately improve patient prognosis.

This study identified 18 OSRDEGs in RM, highlighting the intricate relationship between OS and pregnancy mainte-nance. Through intersectional analysis, six hub genes (*ARRB2*, *BMF*, *SORCS2*, *STK3*, *UCN2*, and *VIPR1*) were iden-tified. Beta-arrestin 2 (*ARRB2*) is a versatile protein that modulates G protein-coupled receptor signaling and affects various cellular processes, such as receptor desensitization, internalization, and downstream signaling pathways [35]. *ARRB2* may regulate inflammatory responses by influencing the infiltration of immune cells, such as macrophages, neutrophils, and T cells [36]. RM is closely associated with the dysregulation and infiltration of immune cells at the maternal–fetal interface. Specifically, an increase in proinflammatory Th17 cells and a decrease in regulatory T cells represent a core mechanism disrupting immune tolerance [37]. Additionally, the polarization of macrophages toward the proinflammatory M1 phenotype and the formation of neutrophil extracellular traps (NETs) by neutrophils exacerbate local inflammation and thrombosis, collectively creating a proinflammatory and prothrombotic microenvironment unfa-vorable for pregnancy [38]. Therefore, *ARRB2* may play a significant role in the inflammatory regulation of RM by mod-ulating the recruitment or functional states of these immune cells. As a member of the BH3-only protein family, *BMF* is a key regulator of the mitochondrial apoptotic pathway [39]. It is activated under stress conditions such as hypoxia [40] and endoplasmic reticulum stress [41], and changes in membrane permeability changes by antagonizing antiapoptotic proteins such as Bcl-2, thereby initiating the apoptotic cascade [42]. Excessive apoptosis of trophoblasts or endometrial

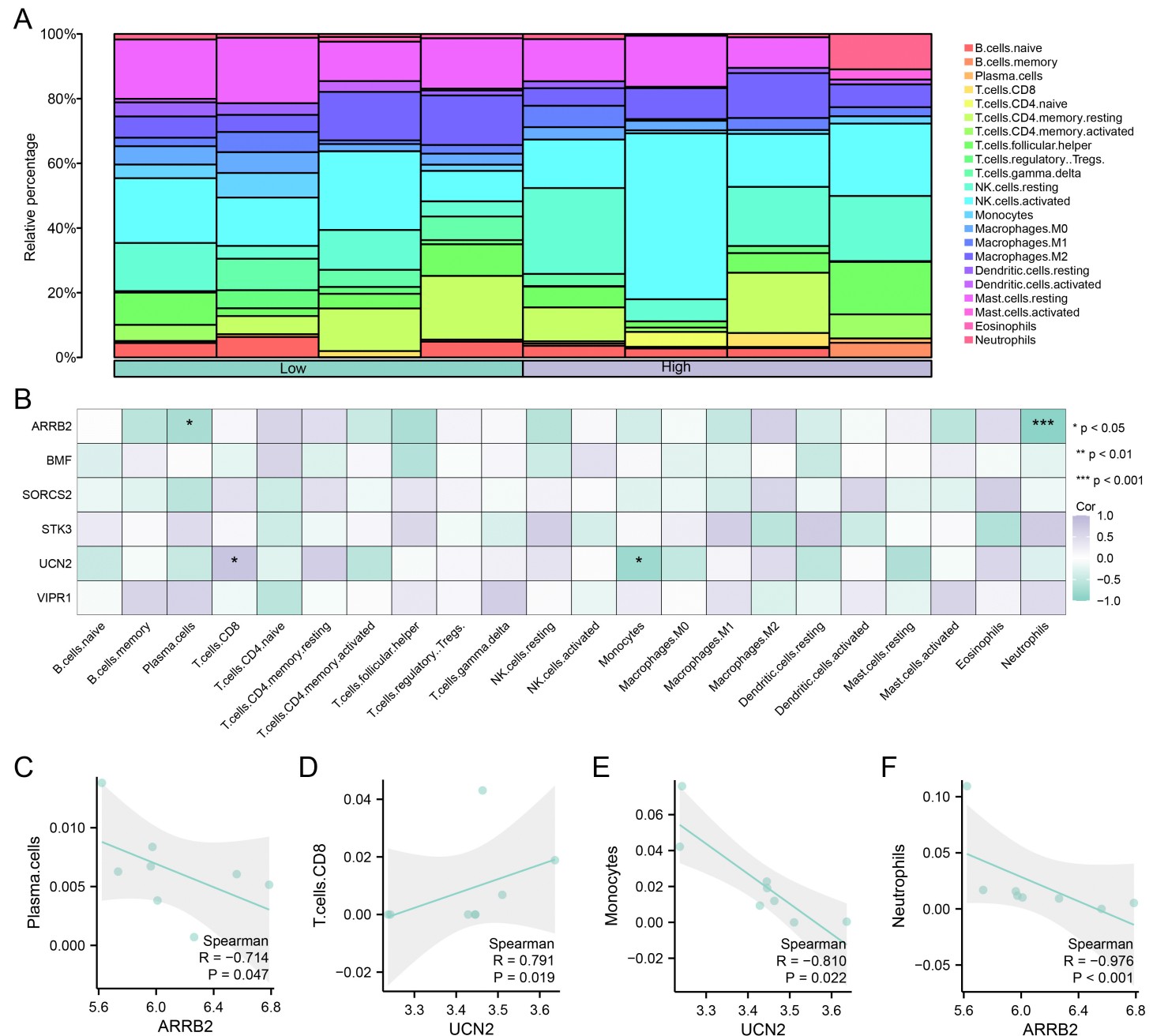

**Fig 11. Analysis of immune system infiltration across the high and low groups in the combined datasets (CIBERSORTx). A.** Stacked bar plot of 22 immune cell types in the high and low groups under the CIBERSORT algorithm. **B.** Correlation heatmap between the infiltration abundance of 22 immune cells and the expression of hub genes. **C-F.** Scatter plots showing significant correlations between hub genes and specific immune cells: (C) *ARRB2* vs Plasma.cells, (D) *UCN2* vs T.cells.CD8, (E) *UCN2* vs Monocytes, (F) *ARRB2* vs Neutrophils. Asterisks in the heatmap (B) indicate statistical significance: ns, P ≥ 0.05 (not significant); *, P < 0.05; **, P < 0.01; ***, P < 0.001.

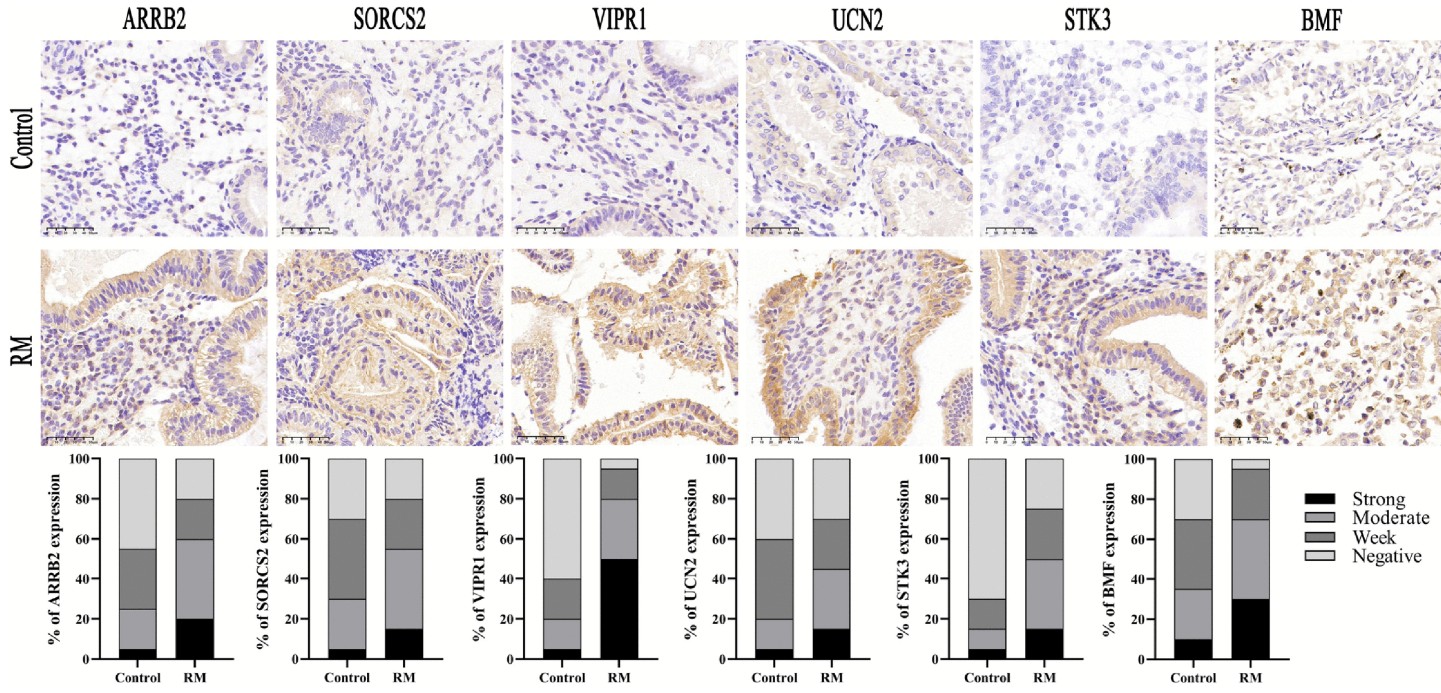

**Fig 12. IHC detection of six proteins in the control and RM groups(scale bar: 50 μm).**

cells is closely associated with gestational disorders such as placental insufficiency and implantation failure [43]. *SORCS2* (sortilin-related VPS10 domain containing receptor 2) is a VPS10P domain receptor that is expressed primarily in the nervous system. It influences neuronal survival and apoptosis by modulating neurotrophic signaling, such as the BDNF/TrkB pathway, and participates in signal transduction cascades, including the MAPK and PI3K/Akt pathways [44,45]. In terms of immunoregulation, *SORCS2* is actively involved in neuroinflammatory processes [46]. This immune enrichment, achieved through the regulation of glial cell activity and the release of inflammatory factors, underpins its role in neurological disorders [47]. Although direct evidence remains limited, the potential function of *SORCS2* in regulating OS, cell migration, and cellular homeostasis at the maternal–fetal interface warrants further investigation. *STK3*, also known as mammalian sterile 20-like kinase 2 (*MST2*), participates in the Hippo signaling pathway, where it functions as a core kinase to regulate cell proliferation, apoptosis, and OS responses, thereby playing a vital role in maintaining homeostasis and development in various tissues [48]. *STK3* regulates cellular proliferation and apoptosis through downstream signaling axes such as the PI3K/AKT axis [49]. Loss of *Mst1/2* function results in increased oxidative damage, phagocyte senescence, and cell death [50]. Dysregulated Hippo pathway activity is closely associated with impaired differentiation, migration, and invasion of trophoblast cells, underscoring its critical role in embryonic implantation and placental development [51,52]. *UCN2*, a member of the corticotropin-releasing factor (CRF) family, plays a significant role in stress responses and immunoregulation [53]. Alterations in *UCN2* expression during immune and cellular stress responses are closely associated with pregnancy outcomes such as preterm birth and preeclampsia [54–56]. *VIPR1* (also known as *VPAC1*) mediates the signaling of vasoactive intestinal peptide (VIP) and is involved in neural, immune, and vascular regulation [57]. At the maternal–fetal interface, VIP modulates immune tolerance, trophoblast function, and vascular development primarily through the VPAC1 and VPAC2 receptors, and deficiency in this signaling pathway can disrupt placental homeostasis, thereby leading to adverse pregnancy outcomes [58]. In summary, the six hub genes (*BMF*, *SORCS2*, *STK3*, *VIPR1*, *UCN2*, and *ARRB2*) operate synergistically within molecular

networks regulating OS, apoptosis, signal transduction, and immune microenvironment dynamics at the maternal–fetal interface. They play central roles in sustaining interface integrity, supporting embryo implantation and development, and modulating inflammatory and immune homeostasis. Dysregulation of these genes may lead to impaired maternal–fetal crosstalk and defective embryonic growth, ultimately promoting RM. This concept is experimentally supported by IHC analysis of endometrial tissues from RM patients, which demonstrated significantly elevated expression of six hub genes compared with controls. These findings provide mechanistic insight into RM pathogenesis and establish a foundation for early risk detection, personalized management, and the development of targeted therapeutic interventions.

Enrichment analysis of the OSRDEGs revealed their involvement in crucial biological processes and pathways. GSEA revealed that the genes distinguishing the RM and control groups in the combined datasets were substantially overrepresented in pathways including hypoxia Dn, Emt Breast Tumor Dn, HCC Progenitor Wnt Up, and Prodrank Tgfb Emt Up. The pathways referenced above, although frequently named for their roles in oncogenesis or specific tissue contexts, in fact underscore the conservation of fundamental molecular mechanisms, including EMT, Wnt signaling, and TGF-β regulation, across a range of physiological and pathological conditions. In the context of pregnancy, EMT is indispensable for trophoblast function during embryo implantation and placental development, as it critically regulates cell migration, invasion, and differentiation [59–61]. Similarly, signaling pathways such as Wnt and TGF-β pathways contribute to trophoblast differentiation, immune modulation, and oxidative stress management [62–64]. Dysregulation of these pathways resulting from aberrant activation or suppression can disrupt trophoblast function, compromise immune tolerance at the maternal–fetal interface, and perturb redox homeostasis, thereby potentially contributing to the pathogenesis of RM. Our GSEA findings support the relevance of these pathways in RM, highlighting their importance in clarifying disease mechanisms and suggesting their potential utility for future therapeutic targeting.

Our analysis investigated the intricate interplay between OS and immune responses in RM. *ARRB2* is significantly negatively correlated with neutrophils, which are implicated in the inflammatory responses triggered by various microbial infections. The *ARRB2*-mediated GPR43-NF-κB signaling pathway might be a plausible target for a variety of inflammatory diseases [65], and the negative regulatory role of *ARRB2* in inflammation has been confirmed by the fact that *ARRB2* deficiency leads to an increase in neutrophils [66]. *ARRB2* acts as a negative regulator of *CXCR2* signaling within the G protein-coupled receptor (GPCR) family [67], and the chemokine receptor *CXCR2* is crucial for neutrophil migration and inflammatory cytokine production [67]. Positive correlation between *UCN2* and T cells.CD8 suggest a potential role for *UCN2* in T-cell subsets. T.cells.CD8 is not only capable of directly inducing cell death [68] but also produces effector cytokines and regulatory molecules in the decidua. RM patients present a considerable increase in decidual T cells, with a focus on T cells. The CD8 population might be associated with the disruption of maternal–fetal immune tolerance [69]. Ran *et al.* [70] reported a transcriptome analysis of RM, revealing abnormal infiltration of immune cells, particularly an abnormal increase in T cells.CD8 and neutrophils in patients with RM. These findings indicate that the gene expression patterns of immune cells could serve as markers of underlying immune dysregulation, leading to a deeper understanding of the etiology of recurrent miscarriages and offering insights into potential therapeutic targets.

On the basis of the constructed mRNA–miRNA regulatory network, *STK3* was identified not only as a signaling hub gene but also as a target under extensive posttranscriptional regulation by multiple miRNAs. Previous studies have confirmed that miRNAs can directly or indirectly modulate the expression of Hippo pathway components, including STK3 [71]. In addition, existing evidence indicates that miRNAs participate in the regulation of trophoblast proliferation, apoptosis, and oxidative stress responses, thereby influencing placental structure and function [72–74]. On the basis of these findings, we hypothesize that in RM, dysregulated interactions between *STK3* and specific miRNAs may concurrently disrupt Hippo signaling and impair trophoblast function, ultimately compromising embryo implantation and placental development, and increasing the risk of miscarriage. *STK3* and its associated miRNAs may represent promising molecular targets for the diagnosis and treatment of RM.

Despite the systematic bioinformatics analyses conducted across multiple cohorts using public databases to identify OSRGs associated with RM and establish a diagnostic model, several limitations warrant consideration. First, the absence of *in vivo or in vitro functional* studies hinders confirmation of the functional roles of the pivotal OSRGs and associated mechanisms in RM, despite the addition of IHC experiments that provided preliminary protein-level validation. Second, Second, while the sample size in both the bioinformatics analysis and IHC experiments was sufficient for generating preliminary data, it remains relatively small and may not fully capture the heterogeneity of the RM population. Third, the absence of prospective clinical studies to validate the diagnostic model's real-world applicability indicates that clinical deployment necessitates rigorous evaluation in larger, multicenter cohorts. Finally, the use of multiple datasets in this study introduces the potential for batch effects, although efforts have been made to address this through statistical adjustments. Future studies will validate and refine the molecular mechanisms and diagnostic model through integrated analysis of clinical samplescombined with in vitro and in vivo experiments, aiming to facilitate the translation of these bioinformatic discoveries into clinical applications.

## Conclusions

In summary, this study identified numerous OSRDEGs associated with RM, with functional enrichment analysis shedding light on BP, CC, and MF and the pathways that these genes might influence. The construction of diagnostic models based on these genes has the potential to improve the prediction and comprehension of RM, which is preliminarily supported by our IHC results from three case–control cohorts. The observed correlations between hub genes and their potential roles in immune cell infiltration offer new insights into the pathophysiology of RM. Further analysis is necessary to ascertain the clinical applicability of this diagnostic model for predicting RM. These findings, including the initial IHC data, provide a foundation for future larger studies and experimental validation to translate bioinformatics discoveries into clinical practice, ultimately enhancing RM management and treatment.

## Supporting information

**S1 Table. List of OSRGs.**
(DOCX)

## Acknowledgments

We would like to thank Editage (www.editage.cn) for English language editing.

## Author contributions

**Conceptualization:** Mian Wang, Xiaoyan Cheng.

**Data curation:** Mian Wang, Lingling Zhu, Xiaoyan Cheng.

**Formal analysis:** Mian Wang.

**Funding acquisition:** Lingling Zhu.

**Investigation:** Mian Wang.

**Supervision:** Xiaoyan Cheng.

**Validation:** Mian Wang, Lingling Zhu.

**Visualization:** Lingling Zhu.

**Writing – original draft:** Mian Wang.

**Writing – review & editing:** Mian Wang, Lingling Zhu, Xiaoyan Cheng.

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
