## [Decision Letter · Decision Letter 0]

30 Apr 2025

Dear Dr. Cheng,

Thank you for submitting your manuscript to PLOS ONE. After careful consideration, we feel that it has merit but does not fully meet PLOS ONE’s publication criteria as it currently stands. Therefore, we invite you to submit a revised version of the manuscript that addresses the points raised during the review process.

We look forward to receiving your revised manuscript.

Kind regards,

Gary S. Stein

Academic Editor

PLOS ONE

Journal Requirements:

“This work was supported by the Municipal Health Commission of Nantong (No. MSZ2023057).”

4. Please note that your Data Availability Statement is currently missing the repository name and/or the DOI/accession number of each dataset OR a direct link to access each database. If your manuscript is accepted for publication, you will be asked to provide these details on a very short timeline. We therefore suggest that you provide this information now, though we will not hold up the peer review process if you are unable.

Reviewers' comments:

Reviewer's Responses to Questions

**Comments to the Author**

1. Is the manuscript technically sound, and do the data support the conclusions?

Reviewer #1: Partly

Reviewer #2: Yes

2. Has the statistical analysis been performed appropriately and rigorously?

Reviewer #1: Yes

Reviewer #2: Yes

3. Have the authors made all data underlying the findings in their manuscript fully available?

Reviewer #1: Yes

Reviewer #2: Yes

4. Is the manuscript presented in an intelligible fashion and written in standard English?

Reviewer #1: Yes

Reviewer #2: Yes

Reviewer #1: This study employed bioinformatics approaches to uncover the differential expression of oxidative stress-responsive genes in RM, and establishes a diagnostic model and provides insights into immune-modulation therapies for RM. However, the study was only based on bioinformatic analyses using publicly available data, and experimental validation on clinical cohorts, cell lines and animal models is recommended. Moreover, the figures in the manuscript are vague and some figure legends are difficult to be recognized, such as figure 10 and figure 11. So the quality of the figures should be improved.

Reviewer #2: This study systematically analyzed the expression characteristics of oxidative stress-related genes in RM by integrating multiple public gene expression datasets using bioinformatics methods, identifying 18 differentially expressed genes and further selecting 6 core genes to construct a high-precision diagnostic model (AUC=0.975). Functional analysis revealed that these genes participate in RM pathogenesis by regulating hypoxia, epithelial-mesenchymal transition, and Wnt signaling pathways, while immune infiltration analysis further demonstrated their significant correlations with neutrophils and CD8+ T cells. Overall, this represents a well-designed and rigorously executed study that offers new perspectives for understanding the molecular mechanisms of RM.

**Do you want your identity to be public for this peer review?** For information about this choice, including consent withdrawal, please see our Privacy Policy

Reviewer #1: No

Reviewer #2: No

---

## [Author Response · Author response to Decision Letter 1]

9 Jun 2025

June 9, 2025

Manuscript ID: PONE-D-25-04005

Title: Identification of oxidative stress-responsive genes in recurrent miscarriage and their role in disease pathogenesis

Dear Dr. Gary S. Stein and Reviewers,

We sincerely appreciate your valuable feedback and constructive comments on our manuscript. We have carefully addressed all points raised by the reviewers and the academic editor. The revised manuscript incorporates all suggested changes, with modifications highlighted in the “Revised Manuscript with Track Changes” file, and detailed responses are provided below. Thank you for your time and expertise in improving our work.

1. Style requirements:

We have reformatted the manuscript according to PLOS ONE templates and ensured consistent file naming.

2. Code sharing guidelines:

Not applicable (this study did not generate new code).

3. Financial disclosure:

This work was supported by the Municipal Health Commission of Nantong (No. MSZ2023057). The funders had no role in study design, data collection and analysis, decision to publish, or preparation of the manuscript.

4. Data availability statement:

All data relevant to the study are included in the article or available as supplemental information. Gene expression data are available in the GEO repository (https://www.ncbi.nlm.nih.gov/geo/) under accession numbers GSE22490, GSE26787, and GSE165004.

5. Supporting information captions:

Captions for supporting files are now included at the end of the manuscript, specific revisions can be found on page 72, lines 917-918.

6. Reference citations and reference list:

We carefully checked the reference citations in the text and the reference list, and removed the extra reference numbers in the main text; for more details, specific revisions can be found on page 24, line 130.

Reference list:

Reference 3: The author list format is incorrect (the abbreviation of RPL EGGo is non-standard), see page 68, line 729.

Reference 5: The page number is incorrect, see page 68, line 737.

Reference 16: The page number is incorrect, see page 69, line 769.

Reference 23: The page number is incorrect, see page 69, line 788.

Reference 28: There are multiple format errors, see page 69, lines 800-801.

Reference 35: The page number is incorrect, see page 70, line 821.

Response to Reviewers’ Comments:

Reviewer #1:

We sincerely appreciate your thorough review and valuable comments.

This study primarily conducted multi-cohort bioinformatic analyses using publicly available databases to identify OSRGs associated with RM and construct a diagnostic model. However, constrained by current resource limitations, experimental validation has not yet been performed. We plan to undertake this essential experimental validation in future work. We have explicitly acknowledged this limitation in the revised manuscript (Page 67, Lines 697-716).

Thank you for highlighting this. We have replaced all low-resolution figures with high-quality versions, with particular focus on optimizing Figures 10 and 11.

Once again, thank you for your valuable suggestions.

Reviewer #2:

We sincerely appreciate your thorough review and valuable comments.

We are particularly encouraged by your positive assessment of our study design, analytical workflow, and result interpretation. Your recognition is highly valued by our team.

Once again, thank you for your valuable suggestions.

Sincerely,

Xiaoyan Cheng.

E-mail: 1012770174@qq.com.

---

## [Decision Letter · Decision Letter 1]

1 Sep 2025

Dear Dr. Cheng,

Thank you for submitting your manuscript to PLOS ONE. After careful consideration, we feel that it has merit but does not fully meet PLOS ONE’s publication criteria as it currently stands. Therefore, we invite you to submit a revised version of the manuscript that addresses the points raised during the review process.

We look forward to receiving your revised manuscript.

Kind regards,

Gary S. Stein

Academic Editor

PLOS ONE

Journal Requirements:

Reviewers' comments:

Reviewer's Responses to Questions

**Comments to the Author**

Reviewer #1: (No Response)

Reviewer #2: All comments have been addressed

Reviewer #3: All comments have been addressed

Reviewer #4: All comments have been addressed

Reviewer #5: (No Response)

2. Is the manuscript technically sound, and do the data support the conclusions?

Reviewer #1: Partly

Reviewer #2: Yes

Reviewer #3: Yes

Reviewer #4: Yes

Reviewer #5: Partly

3. Has the statistical analysis been performed appropriately and rigorously?

Reviewer #1: Yes

Reviewer #2: Yes

Reviewer #3: Yes

Reviewer #4: Yes

Reviewer #5: Yes

4. Have the authors made all data underlying the findings in their manuscript fully available?

Reviewer #1: No

Reviewer #2: Yes

Reviewer #3: Yes

Reviewer #4: Yes

Reviewer #5: Yes

5. Is the manuscript presented in an intelligible fashion and written in standard English?

Reviewer #1: Yes

Reviewer #2: Yes

Reviewer #3: Yes

Reviewer #4: Yes

Reviewer #5: Yes

Reviewer #1: (No Response)

Reviewer #2: I recommend the acceptance of this paper. Strengths: appropriate methods, clear writing, useful findings. Limitations are acknowledged appropriately.

Reviewer #3: (No Response)

Reviewer #4: Minor Revisions: The abstract contains a sentence structure issue: UCN2 was positively associated with T. cells. CD8+ T cells are inversely associated with monocytes. This phrasing is confusing and requires revision for improved clarity (such as: "UCN2 demonstrated positive correlation with CD8+ T cells, which in turn showed negative correlation with monocytes"). Also, verify that gene symbols are formatted in italics consistently across the entire manuscript.

Although the computational analysis demonstrates technical rigor, the findings remain theoretical predictions. The authors' response to the first reviewer's concerns is inadequate. For work submitted as a "Research Article," some degree of experimental confirmation is a reasonable expectation.

I suggest implementing a major revision requirement that includes experimental corroboration of the central discoveries. The basic expectation should involve:

Quantitative RT-PCR verification of the six key genes' expression levels using a separate, small collection of clinical specimens (recurrent miscarriage patients compared to control endometrial or placental samples).

Alternatively, confirmation could be achieved through analysis of an independent publicly accessible dataset not previously utilized in the model development or validation phases.

If experimental validation cannot be provided by the authors, the manuscript would be better positioned for submission to a specialized computational biology publication or should be explicitly recharacterized as predictive research or a methodological contribution.

The fundamental analytical framework demonstrates excellent quality and deserves publication. Nevertheless, the manuscript lacks completeness in its current state. Incorporating experimental validation is essential to elevate these promising computational predictions to findings with biological and clinical significance. The authors should receive the opportunity to supply this validation prior to making a final publication decision.

Reviewer #5: As a reviewer, here's a detailed critique of the article "Identification of oxidative stress-responsive genes in recurrent miscarriage and their role in disease pathogenesis."

This effectively outlines the study's scope and key findings, presenting a compelling case for the research's relevance. However, as with most articles, it has both strong points and areas that would require further clarification in the full manuscript.

1. Strengths

1.1. The research addresses recurrent miscarriage (RM), a major issue in reproductive health, and connects it to oxidative stress, a poorly understood but increasingly relevant factor. The subject matter is highly pertinent and of great interest to the scientific community.

1.2. The abstract clearly states that a bioinformatics approach was used, analyzing data from the Gene Expression Omnibus (GEO). This sets expectations for the type of study and its data-driven nature.

1.3. The findings are presented in a clear and organized manner. The abstract lists key discoveries, including the identification of: 1)18 oxidative stress-responsive differentially expressed genes (OSRDEGs). 2) Six key hub genes (ARRB2, BMF, SORCS2, STK3, UCN2, and VIPR1) proposed as potential biomarkers. 3)Associated biological pathways via GSEA (e.g., hypoxia, EMT). 4) A mRNA-miRNA interaction network, highlighting STK3's central role. 5) Immune cell infiltration correlations, linking specific genes to immune cell types.

1.4. The study goes beyond theoretical findings and directly mentions its potential for developing a diagnostic model and guiding immune-modulation therapies. This highlights the practical value of the research.

2. Weaknesses and Key Questions for the Authors

2.1. Missing Experimental Validation:

This is the most significant limitation. The study is purely computational (bioinformatics). There is no mention of experimental validation using techniques like RT-qPCR, Western blot, or immunohistochemistry to confirm the differential expression of the identified hub genes in a separate cohort of patient samples. Without this, the findings remain hypothetical and require further verification.

2.2. Details on Results and Mechanistic Roles:

The research identifies hub genes, but it offers minimal insight into their potential mechanistic roles in the pathogenesis of RM. For instance, what is the proposed function of STK3 and its interaction with miRNAs in the context of recurrent miscarriage?

Some of the GSEA pathways mentioned (e.g., "EMT in breast tumors," "Wnt signaling in liver cancer progenitors") seem to be directly from the database output rather than being contextualized for RM. The authors should briefly explain in the full paper why these specific pathways are relevant to the disease.

In addition, there are several ambiguous phrases.

The sentence "UCN2 was positively associated with T. cells. CD8+ T cells are inversely associated with monocytes" is slightly ambiguous. A reviewer would wonder if UCN2 is positively associated with both T cells and CD8+ T cells, or if this is a separate finding. The phrasing could be clearer.

Reviewer's Recommendation

Overall, this is a promising study that lays important groundwork for future research. The article is well-structured and presents novel findings.

However, for publication in a reputable journal such as PLOS One, the authors would need to:

1. Include experimental validation of at least a few of the key hub genes (e.g., using patient samples).

2. Expand on the mechanistic roles of the identified genes and their potential contribution to the disease pathophysiology.

3. This study generates excellent hypotheses, but a strong paper would need to show from correlation (bioinformatics) to causation (experimental validation).

**Do you want your identity to be public for this peer review?** For information about this choice, including consent withdrawal, please see our Privacy Policy

Reviewer #1: No

Reviewer #2: No

Reviewer #3: No

Reviewer #4: No

Reviewer #5: **Yes: ** Dr. Abolfazl Akbari (Physiologist)

---

## [Author Response · Author response to Decision Letter 2]

15 Oct 2025

6. Review Comments to the Author

Reviewer #4:

Minor Revisions: The abstract contains a sentence structure issue: UCN2 was positively associated with T. cells. CD8+ T cells are inversely associated with monocytes. This phrasing is confusing and requires revision for improved clarity (such as: "UCN2 demonstrated positive correlation with CD8+ T cells, which in turn showed negative correlation with monocytes"). Also, verify that gene symbols are formatted in italics consistently across the entire manuscript.

Regarding the suggestions for improving sentence structure clarity in the abstract and standardizing the formatting of gene symbols, we have carefully reorganized and refined the relevant content in the revised manuscript to ensure accurate and clear logical relationships.

The sentence structure issue in the abstract has been addressed: “UCN2 was positively associated with T cells” and “CD8+ T cells are inversely associated with monocytes” have been revised accordingly. Specific modifications can be found on page 3, lines 45, 46; page 63, lines 586, 588; page 70, lines 763. 764. 769.

We have carefully reviewed and standardized the formatting of gene symbols throughout the manuscript according to international conventions, applying italic formatting where appropriate. Italic formatting has been applied to gene symbols and related elements at the following specific locations: page 3, lines 36, 42, 44, 45; page 8-24, Table S1, lines 116; page 26-27, Table 2, lines 171; page 33, lines 315, 316, 324; page 41, lines 411, 412, 417; page 42, lines 428, 429, 433, 434, 438, 445, 446; page 44, lines 487, 488, 495; page 48, lines 547; page 49, lines 551; page 49-62, Table 6, lines 560; page 63, lines 585; page 65, lines 630, 631; page 70, lines 757-760.

These modifications ensure consistent and correct formatting in accordance with international scientific writing standards.

Although the computational analysis demonstrates technical rigor, the findings remain theoretical predictions. The authors' response to the first reviewer's concerns is inadequate. For work submitted as a "Research Article," some degree of experimental confirmation is a reasonable expectation.

I suggest implementing a major revision requirement that includes experimental corroboration of the central discoveries. The basic expectation should involve:

Quantitative RT-pCR verification of the six key genes' expression levels using a separate, small collection of clinical specimens (recurrent miscarriage patients compared to control endometrial or placental samples).

Alternatively, confirmation could be achieved through analysis of an independent publicly accessible dataset not previously utilized in the model development or validation phases.

If experimental validation cannot be provided by the authors, the manuscript would be better positioned for submission to a specialized computational biology publication or should be explicitly recharacterized as predictive research or a methodological contribution.

The fundamental analytical framework demonstrates excellent quality and deserves publication. Nevertheless, the manuscript lacks completeness in its current state. Incorporating experimental validation is essential to elevate these promising computational predictions to findings with biological and clinical significance. The authors should receive the opportunity to supply this validation prior to making a final publication decision.

We fully understand the recommendation to validate the six hub genes using qRT-PCR and agree on the importance of experimental verification in enhancing the reliability of the results. Although qRT-PCR provides sensitive and quantitative measurement of transcriptional levels, its application relies heavily on high-quality RNA and well-defined tissue sources. However, early uterine curettage tissues from patients with recurrent miscarriage are often mixtures of endometrial and decidual components, making it difficult to obtain sufficient high-quality RNA for qRT-PCR analysis at this stage. Given these practical limitations, we plan to perform IHC analysis to validate the expression and localization of key proteins. We believe that the IHC results will strongly support our bioinformatic predictions and improve the clinical relevance of our conclusions. The corresponding revisions can be found on page 3, lines 46-49; page 30, lines 238-254; page 31, lines 255, 276-279; page 32, lines 280, 281, 287; page 64, lines 611-615, and Figure 1, Figure 12.

Reviewer #5�Dr. Abolfazl Akbari�:

2. Weaknesses and Key Questions for the Authors

2.1. Missing Experimental Validation:

This is the most significant limitation. The study is purely computational (bioinformatics). There is no mention of experimental validation using techniques like RT-qPCR, Western blot, or immunohistochemistry to confirm the differential expression of the identified hub genes in a separate cohort of patient samples. Without this, the findings remain hypothetical and require further verification.

We fully acknowledge the comment regarding the lack of experimental validation. In response, we will perform additional IHC analysis to validate key findings at the protein level, which will further corroborate the bioinformatics predictions and strengthen the persuasiveness of our study. The corresponding revisions can be found on page 3, lines 46-49; page 30, lines 238-254; page 31, lines 255, 276-279; page 32, lines 280, 281, 287; page 64, lines 611-615, and Figure 1, Figure 12.

2.2. Details on Results and Mechanistic Roles:

The research identifies hub genes, but it offers minimal insight into their potential mechanistic roles in the pathogenesis of RM. For instance, what is the proposed function of STK3 and its interaction with miRNAs in the context of recurrent miscarriage?

Some of the GSEA pathways mentioned (e.g., "EMT in breast tumors," "Wnt signaling in liver cancer progenitors") seem to be directly from the database output rather than being contextualized for RM. The authors should briefly explain in the full paper why these specific pathways are relevant to the disease.

In the revised manuscript, we will further clarify the roles of the hub genes, particularly elaborating on the interaction between STK3 and miRNAs in the pathogenesis of recurrent miscarriage. We will also supplement the biological relevance between the GSEA-enriched pathways and recurrent miscarriage to strengthen the scientific rigor and logical coherence of the findings. The corresponding revisions can be found on page 67, lines 675, 684; page 69, lines 720-744; page 70, lines 745-754; page 71, lines 764-794; page 72, lines 795-802.

In addition, there are several ambiguous phrases.

The sentence "UCN2 was positively associated with T. cells. CD8+ T cells are inversely associated with monocytes" is slightly ambiguous. A reviewer would wonder if UCN2 is positively associated with both T cells and CD8+ T cells, or if this is a separate finding. The phrasing could be clearer.

In response to the comment regarding the need for clearer correlation statements, we will revise the manuscript to more precisely describe the correlations between UCN2 and CD8+ T cells as well as monocytes, in order to avoid any potential ambiguity. Specific modifications can be found on page 3, lines 45, 46; page 63, lines 586, 588; page 70, lines 763. 764. 769.

Reviewer's Recommendation

Overall, this is a promising study that lays important groundwork for future research. The article is well-structured and presents novel findings.

However, for publication in a reputable journal such as PLOS One, the authors would need to:

1. Include experimental validation of at least a few of the key hub genes (e.g., using patient samples).

In accordance with your comment, we will incorporate IHC analysis in the revised manuscript. These experiments will visually confirm the expression and localization of the target proteins in tissue samples, serving as crucial protein-level validation to strengthen the study's conclusions. The corresponding revisions can be found on page 3, lines 46-49; page 30, lines 238-254; page 31, lines 255, 276-279; page 32, lines 280, 281, 287; page 64, lines 611-615, and Figure 1, Figure 12.

2. Expand on the mechanistic roles of the identified genes and their potential contribution to the disease pathophysiology.

Based on your recommendations, we will further elaborate on the molecular mechanisms of the identified hub genes in the revised manuscript and systematically clarify their potential contributions to the pathogenesis of RM, so as to present a more comprehensive perspective on the functional significance of these genes. The corresponding revisions can be found on page 65, lines 634-644; page 66, lines 645-669; page 64, lines 670-694; page 68, lines 695-716.

3. This study generates excellent hypotheses, but a strong paper would need to show from correlation (bioinformatics) to causation (experimental validation).

We will incorporate IHC analysis in the revised manuscript. This experimental approach will visually demonstrate the localization and expression levels of target proteins in tissue samples, thereby providing protein-level confirmation and experimental support for the bioinformatics screening results. These additions will significantly enhance the scientific rigor and persuasiveness of our study conclusions. The corresponding revisions can be found on page 3, lines 46-49; page 30, lines 238-254; page 31, lines 255, 276-279; page 32, lines 280, 281, 287; page 64, lines 611-615, and Figure 1, Figure 12.

The content and language of the initial draft have undergone final optimization on the following pages and lines: page 3, lines 29-31; page 34, line 341; page 72, lines 805-815, 817; page 73, lines 821, 827, 828, 831.

---

## [Decision Letter · Decision Letter 2]

8 Nov 2025

Identification of oxidative stress-responsive genes in recurrent miscarriage and their role in disease pathogenesis

PONE-D-25-04005R2

Dear Dr. Cheng,

We’re pleased to inform you that your manuscript has been judged scientifically suitable for publication and will be formally accepted for publication once it meets all outstanding technical requirements.

Kind regards,

Gary S. Stein

Academic Editor

PLOS ONE

Reviewer's Responses to Questions

**Comments to the Author**

Reviewer #4: All comments have been addressed

Reviewer #5: (No Response)

2. Is the manuscript technically sound, and do the data support the conclusions?

Reviewer #4: Yes

Reviewer #5: (No Response)

3. Has the statistical analysis been performed appropriately and rigorously?

Reviewer #4: Yes

Reviewer #5: (No Response)

4. Have the authors made all data underlying the findings in their manuscript fully available?

Reviewer #4: Yes

Reviewer #5: (No Response)

5. Is the manuscript presented in an intelligible fashion and written in standard English?

Reviewer #4: Yes

Reviewer #5: (No Response)

Reviewer #4: The authors have been very receptive and have made a significant revision, which has made the manuscript very strong. Addition of IHC data is a direct response to the most significant weakness, which will bring the work to a new level of bioinformatic research with proven biological significance. No additional revisions are needed.

Reviewer #5: It should be noted that bioinformatics studies sometimes provide forward-looking opportunities to improve basic studies, however, they cannot be a suitable substitute for them.

**Do you want your identity to be public for this peer review?** For information about this choice, including consent withdrawal, please see our Privacy Policy

Reviewer #4: **Yes: ** Jonah Bawa Adokwe PhD

Reviewer #5: **Yes: ** Dr. Abolfazl Akbari (Physiology)

---

## [Editor Report · Acceptance letter]

PONE-D-25-04005R2

PLOS ONE

Dear Dr. Cheng,

I'm pleased to inform you that your manuscript has been deemed suitable for publication in PLOS ONE. Congratulations! Your manuscript is now being handed over to our production team.

Kind regards,

on behalf of

Dr. Gary S. Stein

Academic Editor

PLOS ONE